# The PeachSNP170K array facilitates insights into a large-scale population relatedness and genetic impacts on citrate content and flowering time
Yaoguang Xu[1,2], Yang Yu[1,2], Xinpeng Qi[1,2], Qi Zhang[1], Jiantao Guan[1], Zhengquan Zhang[1], Jianhua Wei [1] ✉ & Hua Xie[1] ✉

Peach (*Prunus persica*), a model species in the Rosaceae family and a globally significant temperate fruit, requires advanced genotyping tools to accelerate genomics-assisted breeding. To address this need, we developed the PeachSNP170K array and genotyped 489 peach accessions, generating a high-resolution SNP-based kinship framework that surpasses the limitations of traditional pedigree analysis. This approach enabled the identification of genomic regions underlying key phenotypic variations. Genome-wide association studies (GWAS) uncovered 1202 SNPs linked to sugar and acid content, as well as flowering time, including identified loci associated with citrate content and flowering time. Notably, we identified *PpNHX1* (sodium/proton antiporter 1) within a citrate-associated locus, which influences citrate accumulation in peach fruit. Additionally, haplotype analysis revealed a highly selected haplotype, Hap3, within a major flowering-time locus, contributing to low-latitude adaptation. These findings establish the PeachSNP170K array as a powerful tool for high-throughput genomic analysis, providing valuable resources for peach research and breeding.

The development of high-density single-nucleotide polymorphism (SNP) arrays has transformed genomics-assisted breeding (GAB) across various crops. These tools enable high-throughout genotyping, facilitating association studies that enhance our understanding of genetic diversity and agronomic traits in major crops including rice[1], maize[2], and wheat[3], and to fruit crops, such as apple[4–6], pear[7], kiwifruit[8], grapevine[9,10], strawberry[11], and almond[12]. In response to this trend, the International Peach SNP Consortium (IPSC) upgraded the existing peach 9 K SNP array[13] in 2020 by integrating additional 9 K SNPs to fill genomic gaps and incorporate newly identified SNP breeding-relevant markers[14]. These peach arrays have contributed to analyses of genetic diversity analyses and provided breeders with valuable insights into the genetic basis of key agronomic traits[15–19]. Expanding the number of SNPs would enhance genetic evaluations by providing more informative markers, necessitating the development of high-density SNP arrays for improved genomic studies and GAB in peach.

Peach (*Prunus persica* (L.) Batsch, 2n = 2x = 16) is the third most important temperate tree fruit in terms of worldwide production (FAOSTAT 2021; http://faostat.fao.org) with a prized fleshy and palatable mesocarp, and it is an excellent model fruit crop species for the *Rosaceae* family[20,21]. Originating and domesticated in China[22,23], peach spread westward via the ancient Silk Road through Persia (modern-day Iran) and later reached the Americas during the 16th century[24]. Centuries of breeding have improved yield, fruit quality, adaptation, and disease resistance[25–27], resulting in thousands of varieties worldwide[28]. Since the release of the first peach genome assembly (Lovell) in 2013[29], extensive genomic resources, including high-quality reference genomes[30,31] and transcriptomic datasets (*e.g.* Peach Genomic and Transcriptomic Resources: https://www.rosaceae.org/organism/24333?pane=bio_data_1_rsc_genomes) have facilitated genomic studies of both cultivated peaches and their wild relatives, advancing our knowledge of peach domestication and improvement[15,23,31,32].

Over the past decade, genomic research has significantly enhanced our understanding of peach population structure and accelerated the discovery of favorable genetic loci or genes through genome-wide association studies (GWAS)[14,31,33]. However, large-scale assessments of genetic relatedness

[1]Institute of Biotechnology, Beijing Academy of Agriculture and Forestry Sciences/Beijing Key Laboratory of Agricultural Genetic Resources and Biotechnology/Beijing Key Laboratory of Crop Molecular Design and Intelligent Breeding, Beijing, 100097, China. [2]These authors contributed equally: Yaoguang Xu, Yang Yu, Xinpeng Qi. ✉e-mail: weijianhua@baafs.net.cn; xiehua@baafs.net.cn

**Read alignment**

314.16 Gb raw reads from

23 eastern cultivars;

14 western cultivars;

59 Chinese landraces.

**SNP calling and filtering**

1,828,513 raw SNPs called

SNP filtering criteria:

1) missing rate ≤ 0.4;

2) sequencing depth ≥ 2.0;

3) genotype quality ≥ 10;

4) allele quality ≥ 50.

1,126,403 candidate SNPs

**SNP selection**

1) accessions presented ≥ 2;

2) removing of potential repetitive probes;

3) Affymetrix evaluation categories of

"recommended" and "neutral".

620,259 selected SNPs (intermediate SNP array)

**SNP validation and integration**

Validation using 192 accessions:

1) SNPs belongs to "PolyHighResolution"

and "NoMinorHom";

2) SNP call rate ≥ 97.5%;

3) Integration of IPSC 9K array (7,509 SNPs).

173,925 final SNPs

**PeachSNP170K array**

**Fig. 1 | Workflow for developing of the PeachSNP170K array.** This diagram presents the criteria for SNP variant filtering, selection, validation, and integration.

among accessions remains limited, hindering the full exploitation of the genetic diversity in breeding programs. Global peach germplasm collections often include highly related accessions due to intensive inbreeding[34]. For example, 'Chinese Cling', a progenitor of many modern peach cultivars[35], has contributed to significant genetic similarity across breeding programs. Traditional pedigree-based analyses struggle to capture this complexity, whereas high-density SNP arrays offer a powerful alternative by enabling precise SNP-based measures of relatedness. These genomic tools facilitate efficient genetic resource management and maximize genetic gains in breeding.

Before the advent of SNP arrays and GWAS analysis, genomics-assisted breeding was not feasible. With large-scale population studies, GWAS has enabled the identification of key genetic loci and genes in

peach. Several agronomic traits—including fruit shape[36], flesh color[37], texture[38], and fructose content[31]—have been extensively studied. For peach, fruit citrate content and flowering time are very important for peach breeding, influencing fruit flavor and adaptation, respectively. The *D* locus on chromosome 5 has been linked to fruit acidity-related traits through quantitative trait loci (QTL) mapping and GWAS analyses[31,33,39–43]. Candidate genes for the *D* locus and associated molecular markers have been integrated into breeding programs[33,44–47], alongside additional loci on LG1, LG3, and LG7 associated with citrate content[40,41,44]. Flowering time is crucial for the survival and reproduction in peach. Several loci and genes regulating bud dormancy and flowering time have been identified[18,48,49], including the six tandemly repeated *Dormancy-Associated MADS-box* (*DAM 1-6*) genes and their regulators[50–53]. A deeper understanding of these traits is essential for identifying beneficial alleles that can enhance breeding strategies for the improved peach fruit flavor and adaptation.

Here, we present the development and validation of the PeachSNP170K, a high-density SNP genotyping tool designed for large-scale peach population studies. By genotyping 489 accessions, we established SNP-based kinship clusters and identified shared genomic regions within each cluster. Multi-year GWAS analyses identified genetic loci associated with 11 quality- and adaptation-related traits. Biochemical analyses of candidate genes revealed that *PpNHX1* (Na$^+$/H$^+$ antiporter 1), located within a citrate-associated locus on chromosome 8, plays a key role in citrate accumulation in peach mesocarp tissue. Additionally, haplotype analysis of a major flowering-time locus on chromosome 1 identified Hap3 as a strongly selected haplotype, likely contributing to peach adaptation in low-latitude environments. Collectively, our findings provide a robust genomics tool for high-throughput peach research, enhance our understanding of genetic relatedness, and lay the foundation for future breeding improvements.

## Results
### Development of the PeachSNP170K array
To develop a high-density SNP array representing the most significant SNPs across peach genomes, a total of 314.16 Gb of raw sequence data from 96 diverse peach accessions (Supplementary Data 1) were mapped to the Lovell reference genome. These accessions included 59 peach landraces and 37 cultivars from 27 regions across eastern and western countries. An initial set of 1,828,513 SNPs was identified and filtered as described in the "Methods" section, yielding 1,126,403 high-quality SNPs (Fig. 1).

From this pool, 926,790 SNPs present in at least two peach accessions were selected. After removing 30,837 SNPs unsuitable as probes for the Affymetrix Axiom technology due to sequence repetition, 895,953 SNPs remained. These were further filtered based on the *p*-convert value calculated using the Affymetrix Axiom myDesign pipeline, reducing the selection to 620,259 SNPs. This set was used to create an intermediate SNP array that served as the foundation for further development.

Next, 192 peach accessions (Supplementary Data 2) were genotyped using the intermediate SNP array to evaluate its performance. Only the SNPs from the two highest-priority categories—"PolyHighResolution" (98,884 SNPs) and "NoMinorHorm" (90,684 SNPs) (Supplementary Fig. 1)—were retained. SNPs with a call rate below 97.5% were removed, leaving 166,416 SNPs. Additionally, 8995 SNPs from the IPSC peach 9 K SNP array[13] were integrated, while 1486 overlapping SNPs were excluded. This progress resulted in a final set of 173,925 SNP sites for constructing the PeachSNP170K array (Fig. 1; Supplementary Table 1).

### Distribution and validation of the PeachSNP170K array
PeachSNP170K array SNPs were evenly distributed across eight chromosomes (Fig. 2a) with an average interval length of 1307 bp (Supplementary Table 2). Over 90% of the SNPs had at least one neighboring SNP site within ~3 kb (Fig. 2b). Amongst these SNPs, 61,924 (35.61%) were located within the genic regions of 17,367 genes, while 36,348 (20.90%) were within 1 kb upstream or downstream of genes. In comparison, the peach 9 K SNP array

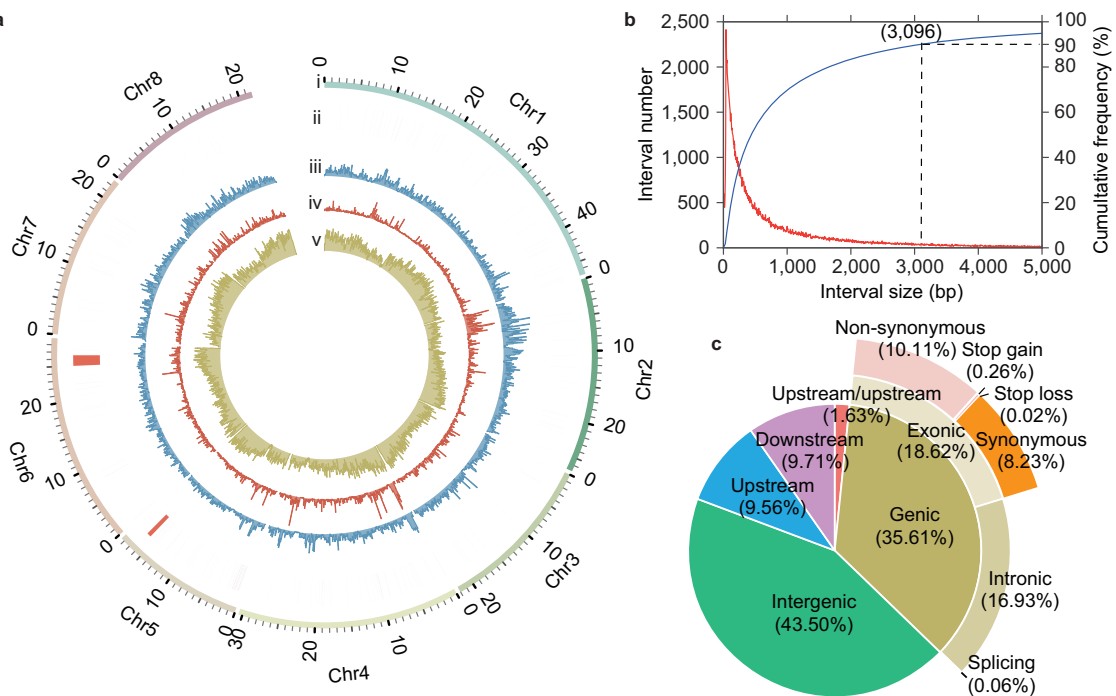

**Fig. 2 | Features of the PeachSNP170K array. a** The distribution of SNPs across the peach genome: (i) all eight peach chromosomes, (ii) publicly available QTLs and associated SNPs (Supplementary Data 10), (iii) SNP density in a 100 kb window for the PeachSNP170K array, (iv) SNP density for IPSC peach 9 K SNP array, and (v) gene density in a 100 kb window. **b** Physical interval distribution between SNP sites on the PeachSNP170K array, showing that over 90% of the intervals are below 3,096 bp. Red curve: interval number; blue curve: accumulated percentage of interval number. **c** Categorization of SNPs into genic, intergenic, upstream, downstream, and upstream/downstream regions, based on reference genome annotation.

contained 8644 (96.09%) and 241 (2.68%) SNPs in these regions, respectively (Fig. 2c; Supplementary Table 3).

Of the SNPs in the coding regions, 17,591 (10.11%) were non-synonymous, 14,308 (8.23%) were synonymous, and 479 were classified as large-effect SNPs (e.g., stop loss and stop gain) (Supplementary Table 3). To assess genotyping accuracy of SNP, re-sequencing data from 50 accessions were analyzed, revealing an average genotype consistency of 93.22% between sequencing and the PeachSNP170 array. This array also exhibited a lower proportion of missing data (0.39%) compared to sequencing (6.71%) (Supplementary Data 3), similar to the Axiom 60 K almond SNP array (0.4–2.7%)[12] and lower than the recently developed 135 K SNP array for kiwifruit (2.5%)[8].

Furthermore, SNPs from the PeachSNP170K array have been anchored onto both the Lovell (v2.0) peach reference genome and the newly released Longhua Shui Mi genome[31], ensuring compatibility and enabling coordinate conversion among various genomes.

### Constructing genetic relatedness for 489 peach accessions using the PeachSNP170K array

Although several large-scale peach population re-sequencing projects have been published[23,26,31,54], none has extensively explored genetic relatedness—such as pedigree information and kinship coefficients—which are crucial for optimizing genetic gain in breeding through the precise genotype selection. Here, we utilized the PeachSNP170K array to genotype 489 diverse peach accessions, encompassing a broad spectrum of cultivars and landraces from 31 regions across 13 countries (Supplementary Data 4). All samples passed quality control, with 151,292 SNPs (86.99%) meeting the quality standards; among these, 132,776 SNPs (76.34%) were classified as "PolyHighResolution", the most stringent category (Supplementary Table 4). Notably, 303 out of the 489 accessions have documented pedigree records involving 642 related accessions.

To gain insight into genetic relatedness, we first constructed a pedigree-based kinship network (Fig. 3a). However, reliance on pedigree records introduces potential inaccuracies due to unobserved relationships, missing data, and errors arising from factors such as pollen contamination or administrative mistakes (e.g., mislabeling and typos). These limitations can compromise the accuracy of genetic relatedness assessments and, consequently, the efficacy of GAB.

To establish a more robust reference for the genetic relatedness, we calculated SNP-based kinship coefficients for all pairwise comparisons. Our analysis revealed a significant correlation ($r = 0.60$, $p < 0.001$) between the SNP-based and pedigree-based kinship coefficients (calculated from 445 known pedigree records) (Fig. 3b). This finding underscores the reliability of SNP-based methods in quantifying genetic relationships and improving pedigree-based records, particularly in cases with incomplete or inconsistent documentation. By offering a genome-wide view of relatedness, SNP-based kinship provides additional insights beyond traditional pedigree data.

For example, a notably high kinship coefficient of 0.86 was observed between 'Bai Hua Shui Mi' and 'Hakuto' (Fig. 3c), reflecting their shared lineage from 'Chinese Cling'. Additionally, within a subset of 'Bai Hua Shui Mi'-derived cultivars, we identified multiple familial relationships, including 13 parent-child pairs, 5 grandparent-grandchild pairs, 19 half-sibling pairs, and 1 full-sibling pair, with the average kinship coefficients of 0.43, 0.31, 0.29, and 0.63, respectively. These results demonstrate how SNP-based kinship analysis enhances genetic insights beyond traditional pedigree data, enabling more precise breeding decisions and genetic management.

To further investigate genetic relatedness and inbreeding within cultivated peach, including both landraces and cultivars, we analyzed inbreeding coefficients (F-values) (Supplementary Table 5). Landraces exhibited a relatively high inbreeding level (F = 0.345), likely due to self-compatibility and long-term selection progresses. In contrast, cultivars displayed lower F-values (mean F: -0.068 to 0.111), reflecting genetic diversity introduced through natural and artificial hybridization, as well as modern breeding practices promoting greater gene flow.

### Kinship clustering among peach accessions

The relationships among peach accessions were visualized using Cytoscape, with kinship coefficients exceeding a threshold of 0.45[55]. Of

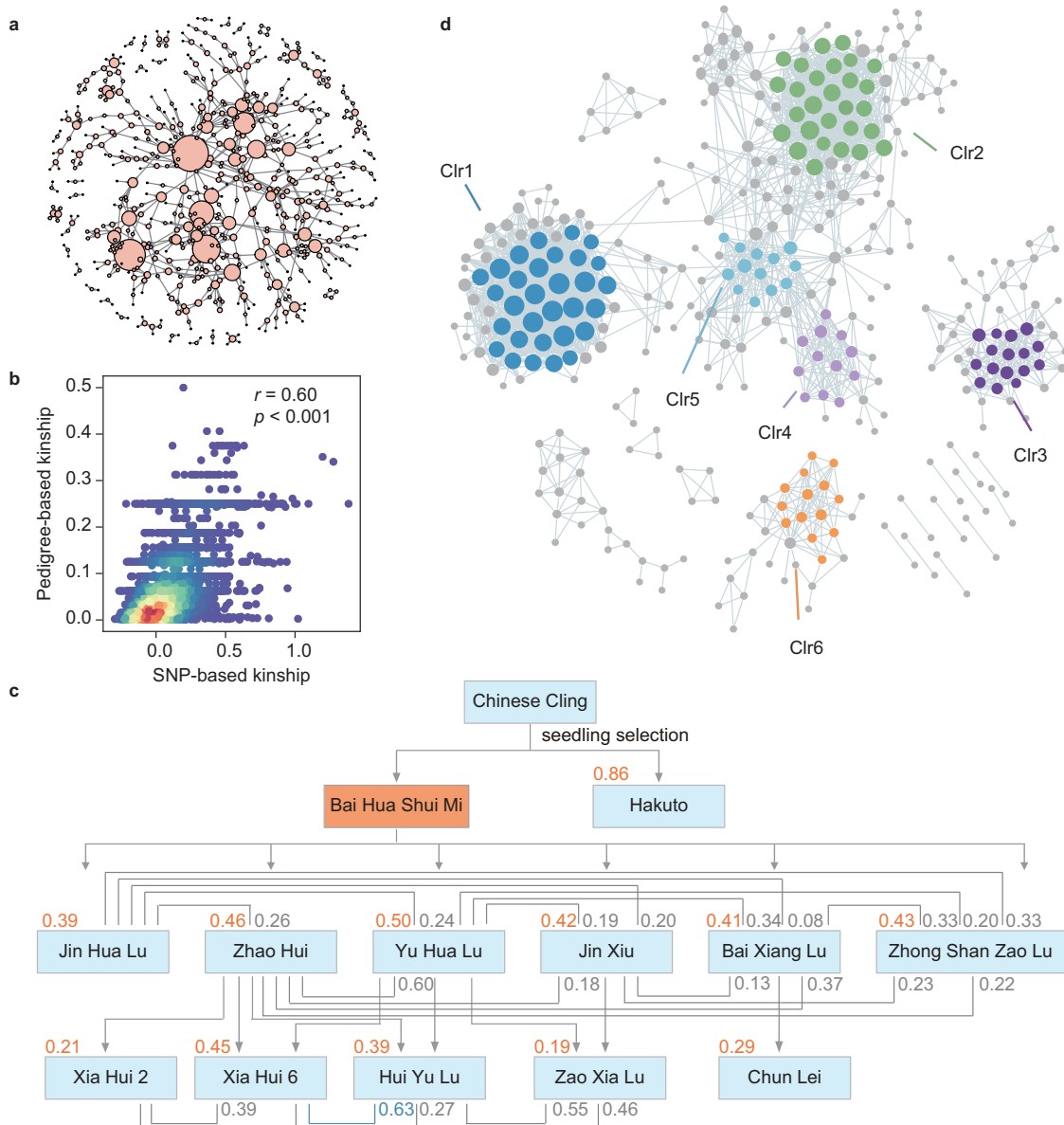

**Fig. 3 | Kinship relatedness and clusters of 489 peach accessions. a** The pedigree network of 642 peach accessions with each node representing an accession and arrows indicating parental-offspring relationships. Node size reflects connectivity. **b** Relationship between pedigree-based and SNP-based kinship coefficients (Pearson's coefficient $r = 0.60$, $p < 0.001$), with blue-to-red color coding density. **c** Kinship coefficients between 'Bai Hua Shui Mi' and its related cultivars, with values indicated in orange (specific accession) and blue/gray (full/half-siblings). **d** Kinship clustering (Clr1–6) based on kinship coefficient greater than 0.45.

the 489 peach accessions analyzed, 382 exhibited kinship coefficients above 0.45 (Supplementary Data 5). Among these, 201 accessions were grouped into 25 multi-member kinship clusters, designated as Clr1 to Clr25 (Fig. 3d, Supplementary Data 6). The remaining accessions consisted of 188 with lower kinship values and 100 forming isolated pairs or small groups that did not meet the clustering criteria. The two largest clusters, Clr1 and Clr2, comprised 59 accessions (29.35%) in total, with average connectivity degrees of 38.00 and 31.07, respectively (Supplementary Data 6). Cultivars such as 'Shenzhou Bai Mi' (47), 'Hakuto' (37), and 'Bai Hua Shui Mi' (37), commonly used in peach breeding programs, were identified within these highly connected clusters (Supplementary Data 6).

The identified kinship clusters provide valuable insights into the molecular bases for relationships among accessions previously unrecorded in pedigree data. For example, Clr3 contains accessions such as 'Sundollar' (P281), 'Sunon' (P169), 'Vallegrande' (P205), 'Early Grand' (P251), 'Taiwan Shui Mi' (P291), 'Chimarrita' (P335), and 'Chirva' (P322), which, despite

lacking documented pedigree links, exhibited high kinship coefficients with other Clr3 members (Supplementary Fig. 2, Supplementary Data 7). The SNP-based kinship analysis proved essential in discerning accessions widely used in past breeding programs. Overall, this approach serves as a valuable genetic reference for selecting parental material in ongoing breeding programs.

**Distinct phenotypes across kinship clusters**

Fruit taste in peaches is largely determined by soluble sugars and organic acid composition. The contents of major soluble sugars (sucrose, glucose, fructose, and sorbitol) and organic acids (malate and citrate) were analyzed across the 489 peaches, along with minor compounds affecting fruit taste, such as quinate and shikimate[56,57]. On average, the sugar proportions were 22.37: 3.63: 2.76: 1 (sucrose:glucose:fructose:sorbitol), while malate-to-citrate ratios ranged from 0.23 to 2.53 (Fig. 4a). Spearman correlation analysis revealed significant relationships among acidity- and sweetness-related traits (Fig. 4b). Titratable acidity (TA) correlated positively with

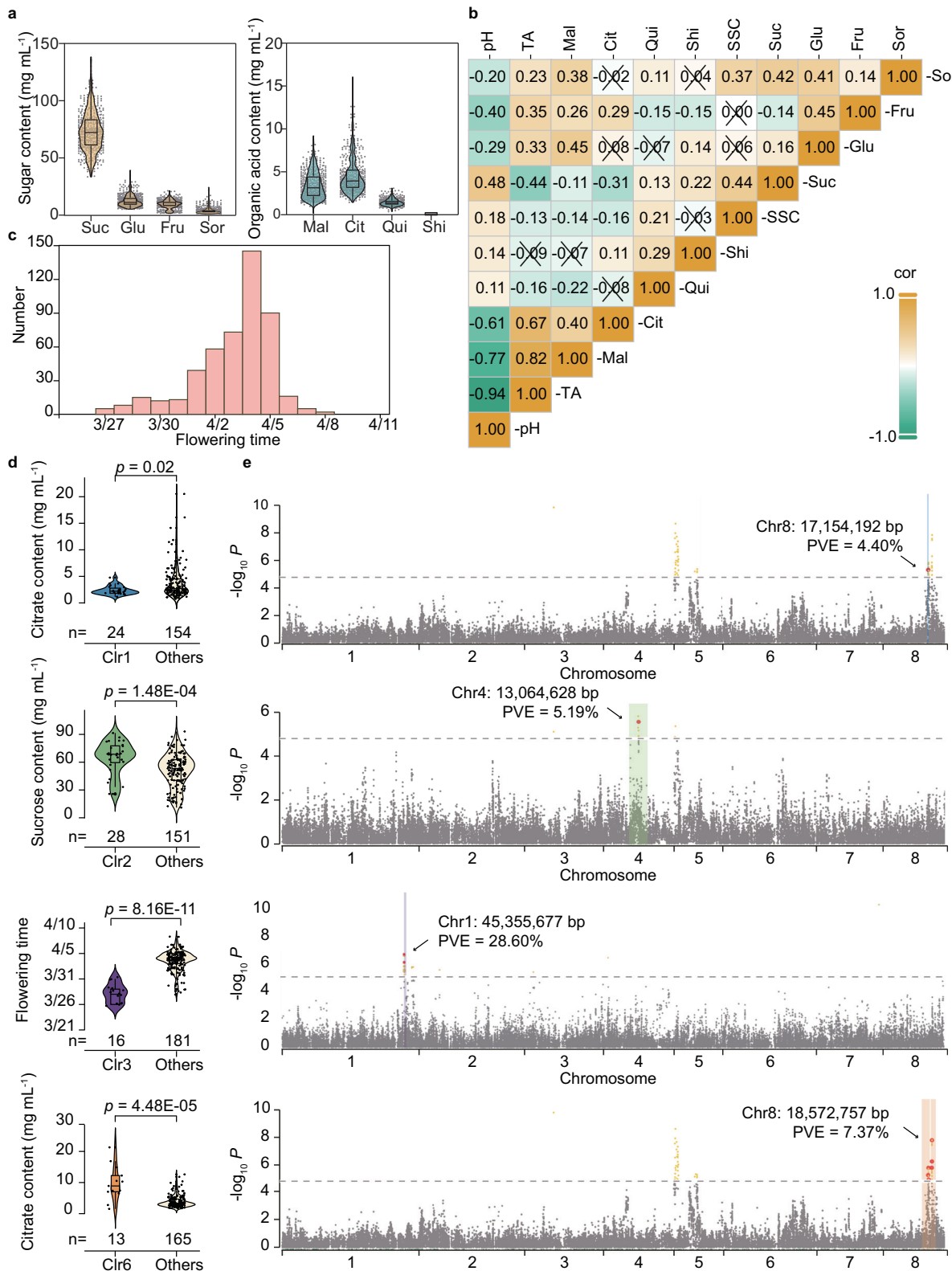

**Fig. 4 | Phenotypes and related GWAS loci located in the shared IBD regions of the clustered peach accessions. a** The contents of major soluble sugar and organic acid components in peach fruit. Suc, sucrose; Glu, glucose; Fru, fructose; Sor, sorbitol; Mal, malate; Cit, citrate; Qui, quinate; Shi, shikimate. **b** Correlation analysis of fruit taste traits. In the table, an "X" denotes a $p \geq 0.05$, indicating that the correlation is not statistically significant as determined by the Spearman rank method. SSC, soluble solid content; TA, titratable acid. In the violin plots, central line: median values; bounds of the box: 25th and 75th percentiles; whiskers: 1.5 x IQR (the interquartile range between the 25th and 75th percentiles). **c** Distribution of flowering time among the 489 peach accessions. **d** Phenotypes of the accessions in Clr1, Clr2, Clr3, and Clr6. Data for fruit flavor phenotypes measured in 2016 and for flowering time recorded in 2019 are shown. The significance of differences was tested by Student's $t$-test if the phenotypic data followed a normal distribution, or the Mann-Whitney U test if not. **e** GWAS loci associated with the phenotypes located in the IBD regions shared by the accessions of Clr1, Clr2, Clr3, and Clr6. The peak-associated SNPs are colored in red, and their PVE values are shown.

malate ($r = 0.82$) and citrate ($r = 0.67$), and negatively with pH ($r = -0.94$). Malate and citrate were positively correlated ($r = 0.40$). Soluble solid content (SSC) correlated positively with sucrose ($r = 0.44$) and sorbitol ($r = 0.37$), while glucose and fructose were positively correlated ($r = 0.45$). Notably, TA correlated positively with glucose ($r = 0.33$) and fructose ($r = 0.35$), whereas SSC showed a significant negative correlation with malate ($r = -0.14$) and citrate ($r = -0.16$). These results suggest that sweeter peaches tend to have lower acidity, highlighting a complex interplay between acids and sugars in shaping peach flavor profiles.

Flowering time was also examined, with observed variation spanning about 15 days. More than half of the accessions flowered between April 2$^{nd}$ and April 5$^{th}$, influenced by local temperature conditions. On the peak flowering day, approximately 30.15% of the trees were in bloom (Fig. 4c). This distribution suggests that, despite some variation, most accessions exhibited flowering within a relatively narrow timeframe, offering insights into the global flowering patterns of peach trees.

Significant phenotypic differences were observed across kinship clusters. Clr1, consisting primarily of landraces from Northern China, displayed low citrate content ($p = 0.02$) (Fig. 4d), along with unimproved traits such as stony-hard flesh, high quinate content ($p = 1.50E-06$), and elevated shikimate levels ($p = 9.26E-05$) (Supplementary Fig. 3). In contrast, Clr2, closely resembling Chinese Cling peaches, exhibited high sucrose content ($p = 1.48E-04$), increased pH ($p < 2.22E-16$), and reduced malate content ($p = 2.99E-06$) (Fig. 4d, Supplementary Fig. 3).

Clr3, predominantly consisting of accessions from low-latitude regions such as Florida (USA) and Brazil, was characterized by an early flowering phenotype ($p = 8.16E-11$) (Fig. 4c, Supplementary Fig. 3). Notably, most Clr3 and Clr6, which primarily include western countries, exhibited high fruit acidity. Clr3 accessions demonstrated significantly lower pH ($p = 5.80E-04$) and elevated citrate content ($p = 4.58E-03$), while Clr6 accessions displayed low pH ($p = 4.60E-15$) alongside high malate ($p = 1.25E-03$) and citrate ($p = 4.48E-05$) (Fig. 4d, Supplementary Fig. 3). These findings align with previous reports indicating that western cultivars generally possess higher acidity levels compared to eastern cultivars[58].

Further analysis revealed that specific loci associated with these phenotypes were located within the shared identical by descent (IBD) regions for each cluster (Supplementary Data 8). For example, SNPs associated with citrate content were identified within the shared IBD regions of Clr1 and Clr6 accessions (Fig. 4e), which exhibited significantly low and high citrate content, respectively (Fig. 4d). Similarly, SNPs linked to sucrose content and flowering time were found within the shared IBD regions of Clr2 and Clr3 accessions (Fig. 4e), corresponding to their high sucrose levels and early flowering characteristics (Fig. 4d). These findings provide a foundation for the development of molecular markers for trait selection in breeding programs.

## High-density SNP mapping reveals genetic loci for agronomically important traits

Utilizing PeachSNP170K for high-density and high-throughput genotyping is essential for background selection and kinship analysis of related breeding germplasm. Developing genetic markers associated with agronomically important traits is crucial for successful marker-assisted selection (foreground selection). Considering population genetic relatedness, GWAS was conducted using FaST-LMM[59], a reformulated linear mixed model capable of capturing population kinship relationships as a covariate.

Significantly associated loci for fruit flavor-related traits, including sugars (sucrose, fructose, glucose, and sorbitol) and acidity (pH, titratable acid, and contents of malate, citrate, quinate, and shikimate), as well as flowering time, were identified across two consecutive years based on PeachSNP170K array genotyping data for 489 accessions (Supplementary Fig. 4, Supplementary Data 9).

A total of 1202 SNPs associated with these traits were identified (Supplementary Data 9). These SNPs were found within loci significantly correlated with sweetness and acidity components that define the fruit's flavor profile. Additionally, GWAS revealed the peak SNPs within each

linkage disequilibrium (LD) block and calculated their phenotypic variance explanation (PVE) values for each trait (Supplementary Data 9).

For sweetness-related traits, SNPs associated with monosaccharide content exhibited high PVE values, explaining up to 8.45% and 9.10% of phenotypic variation for fructose and glucose content, respectively (Supplementary Fig. 4). This precise mapping underscores the potential of these markers in selective breeding programs aimed at enhancing fruit sweetness.

For fruit acidity, a strong GWAS signal was noted, with a SNP (Chr5: 634,413 bp) significantly associated with all the four acidity-related traits (pH, TA, malate and citrate content), explaining a large proportion of the phenotypic variance (ranging from 7.91% to 29.44%) (Supplementary Fig. 4, Supplementary Data 9). These findings suggest its potential contribution to total acidity. A newly identified signal on chromosome 8 was associated with citrate content, explaining up to 7.37% of the phenotypic variation (Fig. 4e). This locus offers potential avenues for modifying citrate levels in peaches, improving the fruit's flavor profile. Significant SNPs were also identified for quinate and shikimate content, explaining up to 6.13% and 6.17% of phenotypic variation, respectively (Supplementary Fig. 4, Supplementary Data 9).

GWAS analysis of flowering time repeatedly detected a major signal (Chr1: 45,355,677–48,792,036 bp) across two years, explaining up to 30.65–40.63% of the phenotypic variance (Supplementary Data 9).

The PeachSNP170K array not only tags SNPs directly associated with agronomically important traits but also integrates these markers with publicly available quantitative trait loci (QTLs) related for a total of 31 traits (Fig. 2a, Supplementary Data 10). This integration facilitates a more comprehensive breeding approach, combining marker-assisted selection with QTL analysis.

## Genetic loci associated with citrate content and the impact of *PpNHX1* in peach

In our GWAS, a notable signal on Chr8 associated with citrate content was identified (Fig. 5a). Within this region, a linkage disequilibrium (LD) block (Chr8: 16,950–17,180 kb) overlapped with the shared identical-by-descent (IBD) regions of both Clr1 (characterized as low-citrate peaches) and Clr6 accessions (high-citrate peaches), suggesting a genetic basis for citrate accumulation variance (Fig. 5a, Supplementary Data 8).

Within this block, 21 genes were identified. Among these, *Pp.LH.08G01962* (Chr8: 17,217,661-17,222,197 bp) (Fig. 5b), annotated as a sodium/proton antiporter (NHX) gene, was designated *PpNHX1*. Transient overexpression of *PpNHX1* in peach mesocarp tissues resulted in significantly higher citrate contents compared to empty vector controls (Fig. 5c), suggesting its potential role in citrate accumulation. Strong selective signatures (XP-CLR) were observed within the *PpNHX1*-associated genomic block (Chr8: 17,154,192–17,230,631 bp) in Clr1 and Clr6 accessions (Supplementary Fig. 5). This suggests that divergent selection acted on this region, likely reflecting artificial selection preferences for citrate accumulation in these clusters. Such selection may indicate the breeding significance of *PpNHX1* in shaping fruit acidity traits in peach.

Next, the haplotypes of *PpNHX1* were examined. Three SNPs in the genic region of *PpNHX1* formed five haplotypes: Hap1 (74.94%), Hap2 (13.52%), Hap3 (5.96%), Hap4 (4.59%), and Hap5 (0.99%) (Fig. 5b; Supplementary Table 6). The citrate content in accessions carrying Hap2 was significantly higher on average (5.69 mg mL$^{-1}$) compared to those with other haplotypes, whose citrate levels ranged from 1.76 to 3.43 mg mL$^{-1}$ (Fig. 5d; Supplementary Table 6). This variance highlights Hap2's relevance to citrate content, particularly as it is the predominant haplotype in Clr6, known for high-citrate peaches (Fig. 5e). Moreover, the distribution of *PpNHX1* haplotypes between Clr1 and Clr6 suggests a pattern of divergent selection linked to citrate preference, supporting its potential utility in breeding strategies aimed at optimizing fruit acidity.

## Advancing genetic understanding of flowering time and low-latitude adaptation

Flowering time is a complex trait that is of great importance for adapting to different environments[60]. Despite extensive research in model organisms

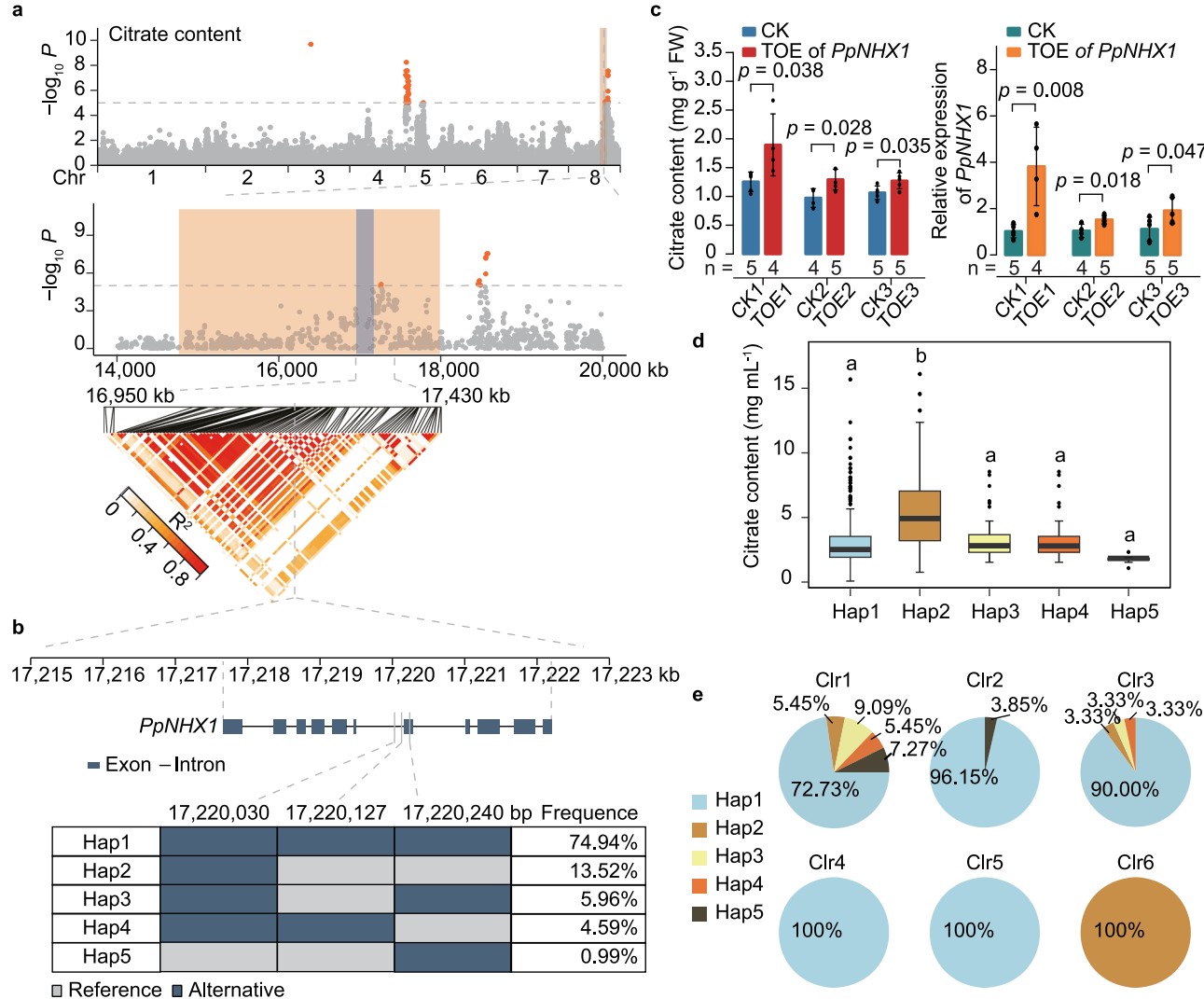

**Fig. 5 | *PpNHX1* contributes to the differences in citrate accumulation between Clr1 and Clr6 accessions. a** A significantly associated LD block of the GWAS locus for citrate content. The shared IBD regions of Clr1 and Clr6 accessions are marked in blue and orange, respectively. Genome-wide (top) and regional (middle) Manhattan plots display GWAS results for citrate content measured in 2016. The horizontal line represents the Bonferroni-adjusted significance threshold. The heatmap of the block (Chr8: 17,154,192-17,230,631 bp), extending 200 kb on each side, is shown with red triangles. **b** Gene structure and the haplotypes of *PpNHX1*. The frequency of each haplotype was calculated based on the 489 peach accessions. **c** Citrate contents in peach mesocarp tissues transiently overexpressing (TOE) *PpNHX1* was significantly higher compared to tissues transiently transformed with the empty vector control (CK). FW, fresh weight. Data are presented as the mean ± SD. Significance was tested with two-sided Student's *t*-tests. **d** Box plot showing the distribution of citrate contents for each haplotype. Multiple comparisons were conducted using the least significant difference test. In the box plots, the central line: median values; bounds of the box: 25th and 75th percentiles; whiskers: 1.5 x IQR (the interquartile range between the 25th and 75th percentiles). **e** The frequency distribution of *PpNHX1* haplotypes (Hap1–5) across six kinship clusters (Clr1–6). The different colors in each pie chart represent the percentage of different haplotypes.

and other crop species[61,62], studies on peach remain limited. We tracked the flowering time of 489 individuals in a peach population, capturing precise flowering dates over two consecutive years. GWAS consistently identified a major signal on Chr1 associated with flowering time across both years (Fig. 6a, Supplementary Fig. 4).

Within this major flowering time signal, haplotype blocks were estimated using PLINK, leading to the identification of a haplotype block (Chr1: 45,355,352-45,435,638 bp) containing two candidate genes (out of 18 protein-coding genes): *Pp.LH.01G06402* and *Pp.LH.01G06403* (Fig. 6a). These genes are orthologs of *AtMED18* (*AT2G22370*)[63] and *AtCRY1* (*AT4G08920*)[64], which are known regulators of flowering time. Notably, this haplotype block is located within the shared IBD regions of Clr3 accessions, primarily from low-latitude regions with significantly early flowering phenotypes (Fig. 6a, Supplementary Fig. 6). Strong selective signatures were detected in this block when comparing Clr3 accessions with other

clusters (Fig. 6b), suggesting that these candidate genes may have contributed to low-latitude adaptation in peaches.

Three haplotypes (Hap1-3) were constructed based on 20 SNPs within this block (Fig. 6c, Supplementary Table 7) after filtering the low-frequency haplotypes (i.e., those present in only one accession). Accessions carrying Hap3 showed significantly earlier flowering times compared to Hap1 and Hap2 accessions (Fig. 6d). Notably, 75.9% of Clr3 accessions, primarily from Florida (USA) and Brazil, carried the Hap3 allele (Fig. 6e), indicating the selection for early flowering in low-latitude environments.

Median-joint network analysis revealed that Hap3 is genetically distinct from other haplotypes (Fig. 6f), showing 95.0% altered SNPs compared to Hap1 and 100.0% compared to Hap2 (Fig. 6c). To facilitate early-flowering germplasm selection, a method using Kompetitive Allele-Specific PCR (KASP) markers was developed based on these 20 SNPs (Supplementary Table 8). Additionally, analysis of wild peach (*Prunus kansuensis*) accessions confirmed that Hap3 follows a different evolutionary trajectory

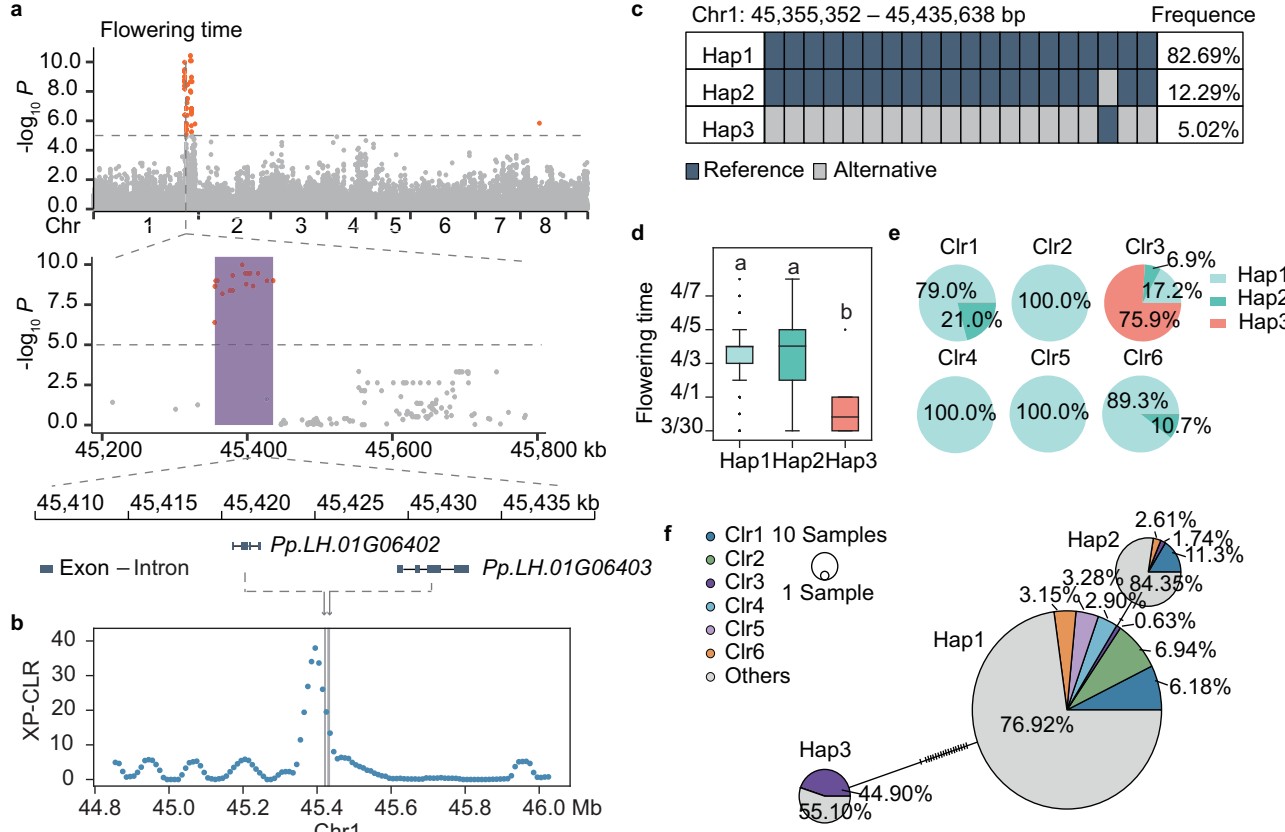

**Fig. 6 | Genetic loci and a low-frequency/distantly evolved haplotype for early flowering in peach. a** Significantly associated loci and candidate genes (*Pp.LH.01G06402* and *Pp.LH.01G06403*) for flowering time. Purple shading represents the IBD regions shared by Clr3 accessions. The horizontal line in the Manhattan plot denotes the Bonferroni-adjusted significance threshold. **b** Selection signature (XP-CLR) comparing Clr3 accessions with those from other clusters. *Pp.LH.01G06402* and *Pp.LH.01G06403*, located within the putative selective regions, are labeled. **c** Haplotypes of the LD block (Chr1: 45,355,352-45,435,638 bp) associated with flowering time. Haplotype frequencies were calculated based on 489 peach accessions. **d** Box plot showing the distributions of the flowering dates among

different haplotypes. Multiple comparisons were conducted using the least significant difference test. In the box plots, central line: median values; bounds of the box: 25th and 75th percentiles; whiskers: 1.5 x IQR (the interquartile range between the 25th and 75th percentiles). **e** The frequency distribution of the three haplotypes associated with peach flowering time across the six kinship clusters (Clr1–6). Different colored sections in each pie chart represent the proportion of each haplotype. **f** Haplotype network of the flowering trait. Median-joint network analysis was used to construct the haplotype network of the flowering haplotypes among the 489 accessions.

than haplotypes found in cultivated peach varieties (Supplementary Fig. 7). Collectively, these findings suggest that Hap3 independently evolved to confer advantages in early flowering under low-latitude conditions, offering valuable insights for breeding programs aimed at enhancing peach adaptability across diverse environments.

## Discussion

Genomics-assisted breeding approaches have significantly enhanced breeding efficiency for major crops[65] and have recently been applied to fruit crops such as apple[6], pear[7], and grapevine[10]. For peach, the IPSC peach 9 K SNP array and an updated 18 K SNP array have been used to genotype diverse accessions and identify genetic loci associated with agronomically important traits[13,15,19,42,66]. However, the resolution power of association studies in peaches has been constrained by the narrow genetic basis resulting from intensive inbreeding relatively low genotyping efficiency. To address this limitation, we developed the PeachSNP170K array using genomic data from a broad spectrum of peach accessions across various geographic regions. This enhanced tool optimizes peach germplasm genotyping, allowing for improved genetic analysis and trait discovery.

Genetic relatedness plays a critical role in peach germplasm breeding[28]. Our study significantly advances the understanding of genetic relatedness in peach germplasm by employing high-resolution SNP-based kinship analysis using the PeachSNP170K array to genotype 489 peach accessions.

Previous research broadly examined peach population structure to trace genomic evolution during domestication and improvement[23,26,31,54]. However, efforts to understand genetic relatedness for QTL mapping have been limited to small-scale studies—one based on pedigrees from seven connected families[67], and another using 215 biallelic markers[34]. Our comprehensive analysis has revealed complex kinship relationships, uncovering previously unrecognized connections, including genetic links between landraces assumed to be unrelated and clarifying the relatedness of descendants from the same progenitor but spread across different breeding programs.

SNP-based kinship analysis provides a genome-wide perspective on genetic relationships and serves as a complementary tool to pedigree data rather than a direct substitute. While our findings demonstrate the utility of SNP-based kinship analysis in assessing genetic relationships, we recognize its limitations, such as the potential impact of Mendelian errors (e.g., IBD = 0) on kinship estimates. Since our study focuses on establishing a genetic reference framework rather than reconstructing pedigrees, these considerations were not incorporated into our analysis.

Our GWAS analysis uncovered the genetic underpinnings of key traits such as fruit acidity and flowering time. For fruit acidity, significant signals were all identified on chromosome 5, where the dominant *D* allele governs the low-acid phenotype. Several candidate genes for the *D* gene have been

reported, and molecular markers for low-acid peaches, such as CPPCT040, have been developed[33,44–47,68]. Additionally, we identified a locus on chromosome 8 associated with citrate content, with a strong candidate gene, *PpNHX1*, which appears to influence citrate accumulation in peach fruit. This finding aligns with research in Arabidopsis, where two *AtNHX* genes regulate vacuolar acidity[69]. Haplotype analysis identified five haplotypes within the *PpNHX1* genic region, with Hap2 accessions (all Clr6 accessions) showing high citrate contents, suggesting that Hap2 may serve as a valuable marker for breeding programs. This comprehensive genetic analysis establishes a robust framework for future breeding efforts, facilitating the selection of high-citrate peaches more effectively through targeted genetic screening.

The integration of selective sweep analysis provides additional insights into the evolutionary and breeding relevance of key loci associated with fruit acidity. Divergent selection between Clr1 (low-citrate content) and Clr6 (high-citrate content) accessions likely reflects artificial selection preferences for citrate accumulation. These findings complement the GWAS results, reinforcing the role of *PpNHX1* as a key determinant of fruit acidity and offering valuable insights for marker-assisted selection strategies targeting this trait.

A major locus on chromosome 1 associated with flowering time was also identified. Candidate genes at this locus included *Pp.LH.01G06402* and *Pp.LH.01G06403*, encoding MED and CRY proteins, respectively. This locus was in close proximity to the well-established flowering time locus *Evergrowing* (*EVG*), which contains six DAM genes[50]. These genes are located within the genomic region Chr1: 46,205,972-46,264,046 bp, approximately 100 kb away from the major loci identified in our study. Flowering time regulation is complex, involving autonomous, photoperiod, vernalization, and gibberellin pathways[70]. In woody plants, it is also linked to bud dormancy and chilling requirements[71,72]. While DAM genes are associated with bud dormancy[51,53,73], the CRY gene likely plays a critical role in thermosensory flowering, as its homolog mediates this pathway in *Arabidopsis thaliana*[74].

Haplotype analysis revealed a low-frequency, early-flowering haplotype (Hap3), which is distantly related to the other haplotypes (Hap1, Hap2) and likely followed a distinct evolutionary trajectory. Given that *Prunus kansuensis* and cultivated peach (*P. persica*) share a recent common ancestor[23]—and that *P. kansuensis* typically flowers earlier than cultivated peaches, we propose that Hap3 represents an ancestral early-flowering haplotype retained from their common ancestor.

In summary, the high-throughput and high-resolution capabilities of our newly developed PeachSNP170K array provide critical insights into peach genetics and substantially enhance breeding strategies. By identifying key genetic determinants of agronomically important traits, our study empowers breeders to improve peach flavor profiles and environmental adaptability, which will undoubtedly facilitate the use of genomics-assisted breeding approaches.

## Methods

### Peach genome data for array design
Genomic data from 96 peach accessions, including landraces and cultivars from diverse geographical locations, were obtained from the Short Read Archive under accession numbers SRA053230[29] and SRA073649[32], as well as from NCBI project PRJNA310042[23] (Supplementary Data 1).

### High-quality SNP identification
A multi-step procedure was employed for SNP detection and quality control. Initially, quality control was performed using fastq_quality_trimmer (http://hannonlab.cshl.edu/fastx_toolkit/index.html) with a minimum Phred quality score of 26 and a minimum length of 80 bp for trimming. High-quality paired-end reads were subsequently mapped to the *P. persica* reference genome (v1.1.a1) (https://www.rosaceae.org/species/prunus_persica/genome_v1.0) using BWA (v0.7.12)[75] with the parameters: "mem -t 4 -k 32 –M". PCR and optical duplicates were removed using SAMtools (v0.1.18)[76].

SNPs were identified from the base quality recalibrated Binary Alignment and Map format (BAM) files using SAMtools with the default parameters. Since the SNP array was developed based on the Lovell (v1.0) reference genome[29], the liftover program (http://genomewiki.ucsc.edu/index.php/Same_species_lift_over_construction) was used to re-coordinate the genotyped SNPs from Lovell (v1.0 to v2.0)[30].

### SNP selection for the intermediate 620 K SNP array
To obtain a list of high-quality variants, SNPs were filtered using the following criteria: (i) missing rate ≤0.4, (ii) sequencing depth ≥2.0, (iii) genotype quality ≥10, and (iv) allele quality ≥50. Among the resulting SNPs, those SNPs present in at least two peach accessions were retained. SNPs with potential sequence repetition issues during probe design were excluded.

For the remaining SNPs, the Affymetrix Axiom myDesign GW bioinformatics pipeline was used to calculate the *p*-convert values for each designed probe. These probes were categorized as "recommended", "neutral", "not recommended", or "not possible". SNP conversion performance was evaluated by considering both the forward and reverse strand directions of the probes provided by the Affymetrix® pipeline. Sequence specificity, binding energies, expected non-specific binding rate, and hybridization to multiple genomic regions were analyzed to generate a *p*-convert value (ranging from 0 to 1), describing the predicted probability of SNP conversion for each probe. The probe set of SNPs categorized as "recommended" and "neutral" was selected for constructing the intermediate 620 K SNP array.

### Final SNP selection for construction of the PeachSNP170K array
Approximately 200 ng of genomic DNA from each accession was used for hybridization to the SNP array following Affymetrix guidelines. Hybridization intensity data were obtained using Affymetrix Genotyping Console software (v.4.2) for genotype calling. Samples were excluded if the Dish quality control value was <0.82 or the call rate was <97.5%.

Affymetrix Gene Chip Command Console Software (AGCC) was used to determine the conversion efficiency of each SNP probe in the array for genotype calling. Hybridization intensity data, clustering, and genotype calling were conducted using Affymetrix Power Tools (v1.15.0) and the R package SNPolisher (v1.5.2)[77]. All SNP variants were classified into six categories: "PolyHighResolution", "MonoHighResolution", "NoMinorHom", "Off-Target Variant", "CallRateBelowThreshold", and "Other". SNPs categorized as "PolyHighResolution" and "NoMinorHom" were selected, filtered (SNP call rate >97.5%), and integrated with SNPs from the IPSC peach 9 K SNP array[13] to form the final SNP set. This set and its flanking sequences were used to produce the PeachSNP170K array.

The distribution of SNPs on the PeachSNP170K array and the IPSC peach 9 K SNP array were visualized using Circos[78]. The coordinates of each SNP on the 170 K SNP array among different genomes (Lovell v1.0, Lovell v2.0, and LHSM) were determined by aligning with 100 bp flanking sequence of the SNP to the respective genomes using BlastN. Functional annotation of SNPs was conducted using ANNOVAR (v2013-06-21)[79], with the Lovell (v2.0) genome serving as the reference.

### Kinship and IBD analysis
Pedigree-based kinship coefficients were calculated using the R package 'kinship2'[80]. The kinship coefficient between the "mutation" and its corresponding cultivar was determined to be 0.97. SNP-based kinship coefficients were calculated for all pairwise comparisons among the 489 peach accessions using GEMMA (v0.98.1-0)[81] with default parameters. A coefficient threshold of 0.45 was applied for kinship clustering. The network image was generated using Cytoscape (v3.6.0)[82]. The Pearson correlation coefficient was calculated to assess the relationship between the SNP-based kinship matrix and the pedigree-based kinship matrices.

Pairwise IBD blocks were identified using Beagle (v5.1) RefinedIBD (v17Jan20.102)[83] with the parameters "length=0.01 lod=2.0". The top 10% most frequent IBD segments shared among peach accessions within each cluster were defined as shared IBD regions.

## Agronomic trait evaluation

Phenotypic data for traits related to fruit flavor, including sugars (SSC, as well as the contents of sucrose, fructose, glucose, and sorbitol) and acidity (pH, TA, and the contents of malate, citrate, quinate, and shikimate), were measured in 2016 and 2017, which was published in our previous study[31]. Briefly, juice from ten ripe peaches per plant was pooled for phenotypic analysis. SSC and pH values were measured using a digital hand-held refractometer (ATAGO Pocket refractometer PAL-1) and a pH electrode (Sartorius, PB-10), respectively. TA was determined by titrating 25 mL of fruit juice with 0.1 mol L$^{-1}$ NaOH to reach a pH of 8.1, following the guidelines of "Fruit and Vegetable Products—Determination of Titratable Acidity" (GB/T 12456, 2008).

For the quantification of organic acid and sugar components, fruit juice was mixed with ethanol (3:7 v/v), centrifuged, and filtered through 0.22-μm PVDF syringe filters before high-performance liquid chromatography (HPLC) analysis on a Shimadzu LC-20A system. Organic acid contents were detected with a photodiode array detector (SPD-M20A) and an InertSustain C18 column (250 mm × 4.6 mm ID, 5 μm, GL Sciences Inc.), eluted with 20 mM monopotassium phosphate (KH$_2$PO$_4$, pH = 2.6) at 40 °C and a flow rate of 1 mL/min, and monitored at 210 nm UV absorbance. Sugar content was analyzed with a refractive index detector (RID-10A) using a Luna® 5 μm NH$_2$ 100 Å column (250 mm × 4.6 mm, Phenomenex), with an 80% acetonitrile mobile phase at 40 °C and a flow rate of 3 mL/min. Calibration curves generated from the external standards were used to quantify organic acids and sugar levels.

For the accurate characterization of flowering time, the flowering date of the 489 accessions was tracked over two consecutive years. In years 2019 and 2020, bud status of each accession was recorded daily at the National Peach Germplasm Repository of China (Beijing). The flowering date for each accession was determined as the date when approximately 10% of the flowers on the tree were in full bloom[67]. The final data were calibrated based on the bud growth status recorded one day before and one day after the initial date.

## GWAS and haplotype analysis

Genotype data were imputed and phased using Beagle v5.1 (v17Jan20.102)[84], yielding a dataset of 61,425 SNPs. A comparison between the genotypes from the re-sequencing project[31] and the PeachSNP170K array was conducted using VCFtools[85].

A linear mixed model was applied using FaST-LMM (v2.06.20130802)[59] for GWAS analysis. The $p$-value threshold for significance was determined as 1/n (Bonferroni correction, with a significance level of 0.05, where n corresponds to the number of SNPs). LD blocks were determined using the R package gpart (v1.0.0)[86] with the parameter "CLQcut = 0.6, hrstType = 'fast'". Haplotypes were constructed using the entire SNP set present in the same LD block, and haplotypes with a frequency of less than two were removed.

## Selective sweeps

XP-CLR was used to detect regions and genes under positive selection. XP-CLR is a method that uses allele frequency differentiation at linked loci between two populations to detect selective sweeps. SNPs with a minor allele frequency (MAF) below 5% were removed from the analysis. Windows containing fewer than five SNPs were excluded from further analysis. Each chromosome was analyzed using XPCLR64 (v1.1) with parameters "-w1 0.0005 200 200 1 -p1 0.9". The average XP-CLR scores were calculated for each 50 kb sliding window with a 10 kb step size. The windows with the highest 1% of XP-CLR scores were identified as candidate selective regions.

## Transient overexpression assay

The full-length ORF of *PpNHX1* was amplified by RT-PCR using gene-specific primers (5′-GAGGACAGCCCAAGCTGAGCTCATGGCTTCC CTCTCAATTGG-3′ and 5′-ACTGGTGATTTCAGCGAATTGGTACCT CAAGAACCAGAAAGGAAAGGAAC-3′). The resulting PCR product was inserted into the pGreen0029 62-SK vector using a Uniclone One Step Seamless Cloning Kit (Genesand, China). The procedure for transient overexpression analysis in peach mesocarps was conducted as described in our previous study[31]. Following infiltration, the peach mesocarps were subjected to citrate content analysis using HPLC and gene expression analysis using qPCR analysis with the *PpNHX1*-specific primers (5′-CCAATTACCGGCCAAAGACTGATT-3′) and (5′-TGCCCACATATCC TATTCCAAACA-3′).

## Development of KASP markers for selection of early-flowering peaches

Based on the flanking sequences of 20 SNPs comprising the early-flowering haplotype (Hap3), a set of KASP markers consisting of three SNPs was developed (Supplementary Table 8). These markers were selected based on phenotypic accuracy, genotyping performance, sample pass rate, and cost-effectiveness, specifically for early-flowering peach selection. If an SNP genotype aligned with the reference allele, it was designated as "R"; otherwise, it was designated as "A" (Alternate). When two or more of the three SNPs exhibited an "R/A" or "A/A" genotype, the sample was recognized as early-flowering.

Analysis revealed that accessions flowering on or before March 31$^{st}$ accounted for 11.0% of the total, defining the early-flowering phenotype. Among the 35 samples identified as early-flowering genotypes, 28 flowered on or before March 31$^{st}$, 3 flowered on April 1$^{st}$, 1 flowered after April 1$^{st}$, and 3 had missing records, yielding an accuracy of 87.5%. In contrast, among the 454 accessions lacking early-flowering genotypes, 25 flowered on or before March 31$^{st}$, 424 flowered on or after April 1$^{st}$, and 5 had missing records, resulting in an accuracy of 94.4%. The overall accuracy of the comprehensive assessment was 94.0%.

## Accession numbers

The cDNA sequence of *PpNHX1* has been submitted to NCBI under number OR713114.

## Statistics and reproducibility

All data are presented in mean ± standard error of the mean, with individual data points displayed in the corresponding boxplots. The significance of differences was assessed using Student's $t$-test when the phenotypic data followed a normal distribution; otherwise, the Mann-Whitney U test was used. For multiple comparisons, the least significant difference test was applied. A $p$-value < 0.05 was considered statistically significant and the exact values were shown. Pearson' correlation analysis was used to assess the relationship between SNP-based and pedigree-based kinship coefficients, while Spearman correlation analysis was used to evaluate the associations among acidity- and sweetness-related traits. Transient overexpression assays and PCR reactions were performed in triplicate. All experiments were conducted with four or more biological replicates, The exact number of replicates used for each experiment is specified in the corresponding figure or Supplementary Data 11.

## Reporting summary

Further information on research design is available in the Nature Portfolio Reporting Summary linked to this article.

## Data availability

All relevant data supporting the findings of this study are provided in the main figures, the Supplementary Information file, and the Supplementary Data files. Source data used for all figures can be found in Supplementary Data 11 and Supplementary Data 12. Additional information is available upon request from the authors.

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

## Acknowledgements

This project was supported by the National Key Research and Development Project of China (Grant no. 2022YFF1003100), the Innovation Capacity Building Foundation (grant no. KJCX20230401), and the Special Program for Innovation (Grant no. KJCX20240408) of the Beijing Academy of Agriculture and Forestry Sciences. Additional support was provided by the National Key Research and Development Project of China (Grant no. 2019YFD1000800), Sponsored by Beijing Nova Program (20230484461), and the Beijing Municipal Science & Technology Project (Z151100001015005), the Innovation Capacity Building Foundation (grant no. KJCX20210432), and the Youth Foundation (grant no. QNJJ202120) from Beijing Academy of Agriculture and Forestry Sciences. The authors acknowledge the germplasm supporting from the National Peach Germplasm Repository of China (Beijing) for germplasm support and the Beijing Academy of Forestry and Pomology Sciences for assistance in sampling. Technical support was provided by Affymetrix, Inc. and CapitalBio Corporation.

## Author contributions

H.X. and J.W. designed and supervised the whole project. Y.X. and Y.Y. drafted the manuscript and conducted data analysis. H.X., Y.X., Y.Y., and X.Q. revised the manuscript. J.G., X.Q., and Q.Z. performed bioinformatics analyses. Y.X. and Y.Y. carried out phenotypic data investigation and

experiments. Z.Z. and Q.Z. contributed to data visualization. All authors participated in manuscript preparation and approved the final version.

## Competing interests

The authors declare no conflicts of interest.
