## [Transparent Peer Review file · Communications Biology]

The PeachSNP170K array facilitates insights into a large-scale population relatedness and genetic impacts on citrate content and flowering time

Corresponding Author: Professor Hua Xie

Version 0:

Reviewer comments:

Reviewer #1

(Remarks to the Author)

This paper details the development of a new peach SNP array, which includes a higher number of SNPs compared to existing arrays. The array was tested in a GWAS study focusing on traits related to sugars, acids, and flowering time. Additionally, the study identifies candidate genes for citrate content and performs a functional validation of the PdNHX1 gene. It also identifies a haplotype linked to flowering time.

The paper is well-written, with a sound strategy and methodology. The results are valuable to the community, and I enjoyed reading it. However, before publication, there are sections that should be improved. The two major concerns are the brief descriptions of the phenotypes (in the abstract, materials and methods, results, and discussion) and the limited discussion of previous knowledge in the bibliography, particularly regarding the analyzed traits.

Major comments:

1. I recommend reinforcing all the aspects related to the description of the phenotypes. For example, in the abstract (line 28) the traits analysed should be specified to allow the reader to understand what the paper is about. Quality and adaption are to general concepts. I would suggest specifying sugar and acid content and flowering time. In the results (line 209), I would add a first paragraph describing the phenotypes (How the different traits are correlated between them? Which are the main sugars and the main acids, What's the global distribution of flowering time in the whole collection...). In the fruit acidity discussion (lines 394-402), a reference to the association found in G5, in the region where the subacid trait associated to gene D has been located and that is being used in peach breeding programs to select subacid individuals, should be add.

2. I will also add more information in the introduction, and discussion about previous knowledge available in the bibliography, especially for peach fruit acidity (references) and flowering time (references). For example, reading the introduction the reader can understand that before SNP arrays and GWAS analysis it was not possible to do genomics-assisted breeding, and most molecular markers used today in peach breeding programs were already available before the first GWAS studies in peach. One example is the subacid trait (D) that was mapped in a biparental population by Dirlwanger et al. 1998 (<https://link.springer.com/article/10.1007/s001220050969>) and molecular markers were described in Eduardo et al. 2014 (<https://link.springer.com/article/10.1007/s11295-014-0789-y>). Another example is the evergrowing gene (Evg), described in Wang et al. 2002 (<https://doi.org/10.1093/jhered/93.5.352>) and that is located in a similar region of G1 where an association has been found with flowering time.

Minor comments:

- Abstract: (line 29) A sentence summarising the results from the GWAS before describing the main results about the PpNHX1 candidate gene for citrate and the discovery of the Hap3 linked to early flowering time, could be useful.

- Introduction:

- line 37: 'substitute' precise by 'high throughput'. Previous tools were also precise.

- line 64: Why the evaluation of genetic relatedness remains inadequate? It is not clear.

- line 77: why citrate is the most important trait for the improvement of peach flavor? I would change 'the most important' by 'very important', or 'fruit citrate content' by 'fruit acidity'.

- M&M – A summary of the procedures used to measure acids and sugars should be included in the text although they have been published elsewhere (line 488).
- line 489: 'differences' must be removed. Traits are measured and in the results they should be analysed and is where it should be stated if they are the same or some differences are identified.
- line 490: 'of the 489 accessions for years'. This sentence should be checked because it looks like something is missing.
- line 493: 'The final data were calibrated...'. Substitute were by was and further explain what calibrating means in the context of this sentence. It is not clear to me the meaning of the sentence.
- Results:
- lines 114-117: how many SNPs in each category?
- line 140: remove 'reference'
- line 160: change 'this accession' by 'these accessions'.
- lines 178-196: Clusters should be better described. Only 201 accessions were clustered. What about the other ones? It is difficult for the reader to know which accession is in which cluster. More information (like name and origin) should be added in Supplementary data 6. Alternatively, cluster information should be moved to Supplementary Data 4.
- line 253: Associations should be clarified. 'We identified 1207 SNPs associated...'. Perhaps this information should be summarized in regions and specifying how many regions associated per trait. Supplementary data 9 should be summarized to make it more easy to understand to the reader.
- line 264: Add a dot after parenthesis and then add 'These findings...'
- line 279: remove 'that'
- line 281: remove 'interestingly'
- line 284: 'within this block'. How many genes are annotated in this block in addition to NHX?
- line 321: change 'alternative' by 'different'
- line 323: I do not think that studies on peach flowering time are limited, and previous knowledge should be summarized here or in the introduction.
- line 328: is this block close to the position where the Evg peach gene has been described (Bielenberg et al. 2004; <https://doi.org/10.1093/jhered/esh057>)?
- line 345: are the 20 SNPs necessary to assess flowering time? Please further explain.

Reviewer #2

(Remarks to the Author)

In this article, Xu et al. report the creation of a high-throughput Axiom genotyping array containing 170K SNPs. The SNPs included in the array were identified in the resequencing of 96 peach accessions through a two-step approach, selecting only "PolyHighRes" and "NoMinorHomo" SNPs in an intermediate array containing 620K SNPs. The SNPs from the array developed by the International Peach SNP Consortium (IPSC) were also included in the final array. Xu et al. use the created array to perform an association study with 11 traits related to quality and adaptation, and to attempt the reconstruction of the relationships of nearly 500 peach accessions from the National Peach Germplasm Repository of China.

The paper appears as clear at a first glance, but several issues remain, especially considering the large amount of results presented. In my modest opinion there is material for at least two papers. In some sections, it is evident that the text was produced by different authors and therefore an effort to harmonize the writing would make the article more readable.

In the presentation of the results of the final array, it is not clear how many of the 170K markers work in the final population and how they are classified into the different classes. Furthermore, it is not clear whether the array will be available through ThermoFisher or other channels or if it will remain private.

The introduction is clear, I only have a couple of minor revisions:

- lines 40-41: the adjective "recently" does not seem appropriate for a reference from 2012 or 2020.
- line 57: the link to the website "rosaceae.org" is too general. This website is a big repository of many different types of data, not only of peaches.

In the results, it is necessary to add details on the performance of the final array. The development of the intermediate array from the resequencing is clearly and appropriately described, but details are missing on the classification of the final SNPs in the 489 accessions used for the association study and for the reconstruction of the parental relationships.

- line 114: verify "MonoHighResolution", in other parts of the article it talks about "NoMinorHom".

Supplementary figure 1: check the plot labels, some of the classes are mislabeled.

- lines 113-117: it is not clear where the total of 173,925 SNPs comes from considering 132,776 + 18,561 + 9K.
- lines 128-129: the reference to PyrSNParray seems inappropriate. Verify if it is a typo otherwise remove it or move the comparison with other arrays to the discussion to avoid confusion.

In the kinship study, it is not clear whether the authors consider the high rate of inbreeding and self-pollination that is present in Prunus?

The use of the kinship value (>0.45) alone, without considering the number of loci that present potential Mendelian errors (IBD = 0), could lead to a very high number of false positives in the identified pedigrees.

In the Methods section and particularly in the subsections "SNP selection for the intermediate 620K SNP array" and "Final SNP selection for construction of the PeachSNP170K array" there is a mix of Results and Methods.

At row 503 there is a section about Selective sweeps but the results of this analysis are not presented and discussed, they are only briefly cited at row 287. The results of this analysis should be better discussed or totally removed from the paper.

Version 1:

Reviewer comments:

Reviewer #1

(Remarks to the Author)

Reviewer #2

(Remarks to the Author)

The authors addressed all the comments from both reviewers and the paper is now suitable for publication.

Reviewers' comments:

Reviewer #1 (Remarks to the Author):

This paper details the development of a new peach SNP array, which includes a higher number of SNPs compared to existing arrays. The array was tested in a GWAS study focusing on traits related to sugars, acids, and flowering time. Additionally, the study identifies candidate genes for citrate content and performs a functional validation of the PdNHX1 gene. It also identifies a haplotype linked to flowering time.

The paper is well-written, with a sound strategy and methodology. The results are valuable to the community, and I enjoyed reading it. However, before publication, there are sections that should be improved. The two major concerns are the brief descriptions of the phenotypes (in the abstract, materials and methods, results, and discussion) and the limited discussion of previous knowledge in the bibliography, particularly regarding the analyzed traits.

Major comments:

1. I recommend reinforcing all the aspects related to the description of the phenotypes. For example, in the abstract (line 28) the traits analysed should be specified to allow the reader to understand what the paper is about. Quality and adaption are to general concepts. I would suggest specifying sugar and acid content and flowering time.

Response: We appreciate your valuable suggestion. Following your guidance, we have revised the abstract and replaced “11 quality and adaptation related traits” with “sugar and acid content, as well as flowering time” (line 28) to improve clarity for the reader. We have also ensured that all phenotype-related descriptions are specified throughout the manuscript.

In the results (line 209), I would add a fist paragraph describing the phenotypes (How the different traits are correlated between them? Which are the main sugars and the main acids, What’s the global distribution of flowering time in the whole collection...).

Response: We sincerely appreciate your insightful comments and suggestions on our manuscript. In response, we have added new paragraphs providing a detailed description of the phenotypes (lines 210-228). Furthermore, we have updated Fig. 4 in the main text to incorporate these aspects, including the major sugar and organic acid components in peach fruits (Fig. 4a), the phenotypic correlation analysis (Fig. 4b), and the distribution of flowering time (Fig. 4c). We believe these additions enhance the clarity and comprehensiveness of our study.

“Results” section, lines 210-228:

“Fruit taste in peaches is largely determined by soluble sugars and organic acid composition. The contents of major soluble sugars (sucrose, glucose, fructose, and sorbitol) and organic acids (malate and citrate) were analyzed across the 489 peaches, along with minor compounds affecting fruit taste, such as quinate and shikimate^{56,57}. On average, the sugar proportions were 22.37: 3.63: 2.76: 1

(sucrose:glucose:fructose:sorbitol), while malate-to-citrate ratios ranged from 0.23 to 2.53 (Fig. 4a). Spearman correlation analysis revealed significant relationships among acidity- and sweetness-related traits (Fig. 4b). Titratable acidity (TA) correlated positively with malate ($r = 0.82$) and citrate ($r = 0.67$), and negatively with pH ($r = -0.94$). Malate and citrate were positively correlated ($r = 0.40$). Soluble solid content (SSC) correlated positively with sucrose ($r = 0.44$) and sorbitol ($r = 0.37$), while glucose and fructose were positively correlated ($r = 0.45$). Notably, TA correlated positively with glucose ($r = 0.33$) and fructose ($r = 0.35$), whereas SSC showed a significant negative correlation with malate ($r = -0.14$) and citrate ($r = -0.17$). These results suggest that sweeter peaches tend to have lower acidity, highlighting a complex interplay between acids and sugars in shaping peach flavor profiles.

Flowering time was also examined, with observed variation spanning about 15 days. More than half of the accessions flowered between April 2nd and April 5th, influenced by local temperature conditions. On the peak flowering day, approximately 30.15% of the trees were in bloom (Fig. 4c). This distribution suggests that, despite some variation, most accessions exhibited flowering within a relatively narrow timeframe, offering insights into the global flowering patterns of peach trees.”

Fig. 4 Phenotypes and related GWAS loci located in the shared IBD regions of the clustered peach accessions. (a) The contents of major soluble sugar and organic acid components in peach fruit. Suc, sucrose; Glu, glucose; Fru, fructose; Sor, sorbitol; Mal, malate; Cit, citrate; Qui, quinate; Shi, shikimate. (b) Correlation analysis of fruit taste traits. In the table, an “X” denotes a $p \geq 0.05$, indicating that the correlation is not statistically significant as determined by the Spearman rank method. SSC, soluble solid content; TA, titratable acid. In the violin plots, central line: median values; bounds of the box: 25th and 75th percentiles; whiskers: 1.5 x IQR (the interquartile range between the 25th and 75th percentiles). (c) Distribution of flowering time among the 489 peach accessions. (d) Phenotypes of the accessions in Clr1, Clr2, Clr3, and Clr6. Data for fruit flavor phenotypes measured in 2016 and for flowering time recorded in 2019 are shown. The significance of differences was tested by Student’s t-test if the phenotypic data followed a normal distribution, or the Mann-Whitney U test if not. (e) GWAS loci associated with the phenotypes located in the IBD regions shared by the accessions of Clr1, Clr2, Clr3, and Clr6. The peak-associated SNPs are colored in red, and their PVE values are shown.

In the fruit acidity discussion (lines 394-402), a reference to the association found in G5, in the region where the subacid trait associated to gene D has been located and that is being used in peach breeding programs to select subacid individuals, should be add.

Response: We appreciate your valuable comments and suggestions. In response, we have added a reference to the association of the *D* locus in the “Discussion” section (lines 419-422) of the revised manuscript.

“Discussion” section, lines 419-422:

“For fruit acidity, significant signals were all identified on chromosome 5, where the dominant *D* allele governs the low-acid phenotype. Several candidate genes for the *D* gene have been reported, and molecular markers for low-acid peaches, such as CPPCT040, have been developed^{33,44-47,68}”

2. I will also add more information in the introduction, and discussion about previous knowledge available in the bibliography, especially for peach fruit acidity (references) and flowering time (references). For example, reading the introduction the reader can understand that before SNP arrays and GWAS analysis it was not possible to do genomics-assisted breeding, and most molecular markers used today in peach breeding programs were already available before the first GWAS studies in peach. One example is the subacid trait (*D*) that was mapped in a biparental population by Dirlwanger et al. 1998 (<https://link.springer.com/article/10.1007/s001220050969>) and molecular markers were described in Eduardo et al. 2014 (<https://link.springer.com/article/10.1007/s11295-014-0789-y>). Another example is the evergrowing gene (*Evg*), described in Wang et al. 2002 (<https://doi.org/10.1093/jhered/93.5.352>) and that is located in a similar region of G1 where an association has been found with flowering time.

Response: Thank you for pointing this out. We have carefully revised the Introduction and Discussion sections to provide a more comprehensive overview of previous studies on peach fruit acidity and flowering time (“Introduction” section: lines 70-72; “Discussion” section: lines 419-422 and lines 436-445).

“Introduction” section, lines 70-72:

“Before the advent of SNP arrays and GWAS analysis, genomics-assisted breeding was not feasible. With large-scale population studies, GWAS has enabled the identification of key genetic loci and genes in peach.”

“Discussion” section, lines 419-422:

“For fruit acidity, significant signals were all identified on chromosome 5, where the dominant *D* allele governs the low-acid phenotype. Several candidate genes for the *D* gene have been reported, and molecular markers for low-acid peaches, such as CPPCT040, have been developed^{33,44-47,68}”

“Discussion” section, lines 436-445:

“Candidate genes at this locus included *Pp.LH.01G06402* and *Pp.LH.01G06403*, encoding MED and CRY proteins, respectively. This locus was in close proximity to the well-established flowering time locus *Evergrowing (EVG)*, which contains six DAM genes⁵⁰. These genes are located within the genomic region Chr1: 46,205,972-46,264,046 bp, approximately 100 kb away from the major loci identified in our study. Flowering time regulation is complex, involving autonomous, photoperiod, vernalization, and gibberellin pathways⁷⁰. In woody plants, it is also linked to bud dormancy and chilling requirements^{71,72}. While DAM genes are associated with bud dormancy^{51,53,73}, the CRY gene likely plays a critical role in thermosensory flowering, as its homolog mediates this pathway in *Arabidopsis thaliana*⁷⁴.”

Your insightful suggestions have significantly enhanced the manuscript by improving the clarity of the theoretical background and strengthening the discussion of prior research on peach fruit acidity and flowering time. These revisions provide a more comprehensive context for our study and better highlight the advancements made possible by SNP arrays and GWAS analysis.

Thank you again for your valuable feedback. I appreciate your guidance in improving the quality of our manuscript.

Minor comments:

- Abstract: (line 29) A sentence summarising the results from the GWAS before describing the main results about the PpNHX1 candidate gene for citrate and the discovery of the Hap3 linked to early flowering time, could be useful.

Response: Thank you for your kind suggestion. We have updated the abstract to include a summary sentence on the GWAS results (lines 27-28).

“Abstract” section, lines 27-28:

“Genome-wide association studies (GWAS) uncovered 1,202 SNPs linked to sugar and acid content, as well as flowering time, including novel loci associated with citrate content and flowering time.”

- Introduction:

- line 37: ‘substitute’ precise by ‘high throughput’. Previous tools were also precise.

Response: We have replaced "precise" with "high-throughout" to more accurately describe the advantage of SNP arrays over earlier tools (line 37).

- line 64: Why the evaluation of genetic relatedness remains inadequate? It is not clear.

Response: Thank you for your insightful comment. We agree that "inadequate" is not the most precise term. We have replaced it with "limited", as current method do not fully capture genetic diversity and population structure in peach germplasm (line 63).

- line 77: why citrate is the most important trait for the improvement of peach flavor? I would change 'the most important' by 'very important', or 'fruit citrate content' by 'fruit acidity'.

Response: Following your guidance, we have revised the sentence to read: "For peach, fruit citrate content and flowering time are very important for peach breeding, influencing fruit flavor and adaptation, respectively" which more accurately conveys the role of citrate in taste (lines 73-74).

- M&M – A summary of the procedures used to measure acids and sugars should be included in the text although they have been published elsewhere (line 488).

Response: A summary of the acid and sugar measurement methods has been incorporated into the "Methods" section (lines 524-538) to provide context for readers unfamiliar with the referenced studies.

"Methods" section, line 524-538:

Briefly, juice from ten ripe peaches per plant was pooled for phenotypic analysis. SSC and pH values were measured using a digital hand-held refractometer (ATAGO Pocket refractometer PAL-1) and a pH electrode (Sartorius, PB-10), respectively. TA was determined by titrating 25 mL of fruit juice with 0.1 mol L⁻¹ NaOH to reach a pH of 8.1, following the guidelines of "Fruit and Vegetable Products—Determination of Titratable Acidity" (GB/T 12456, 2008).

For the quantification of organic acid and sugar components, fruit juice was mixed with ethanol (3:7 v/v), centrifuged, and filtered through 0.22- μ m PVDF syringe filters before high-performance liquid chromatography (HPLC) analysis on a Shimadzu LC-20A system. Organic acid contents were detected with a photodiode array detector (SPD-M20A) and an InertSustain C18 column (250 mm \times 4.6 mm ID, 5 μ m, GL Sciences Inc.), eluted with 20 mM monopotassium phosphate (KH₂PO₄, pH = 2.6) at 40°C and a flow rate of 1 mL/min, and monitored at 210 nm UV absorbance. Sugar content was analyzed with a refractive index detector (RID-10A) using a Luna[®] 5 μ m NH₂ 100 Å column (250 mm \times 4.6 mm, Phenomenex), with an 80% acetonitrile mobile phase at 40°C and a flow rate of 3 mL/min. Calibration curves generated from the external standards were used to quantify organic acids and sugar levels."

- line 489: 'differences' must be removed. Traits are measured and in the results they should be analysed and is where it should be stated if they are the same or some differences are identified.

Response: We have removed the term "differences" to align with appropriate scientific phrasing

(line 539).

- line 490: 'of the 489 accessions for years'. This sentence should be checked because it looks like something is missing.

Response: We have clarified the sentence to read: "For the accurate characterization of flowering time, the flowering date of the 489 accessions was tracked over two consecutive years." (lines 539-540).

- line 493: 'The final data were calibrated...'. Substitute were by was and further explain what calibrating means in the context of this sentence. It is not clear to me the meaning of the sentence.

Response: Thank you for your comment. We have corrected "were" to "was" (line 543) and clarified the calibration process as follows:

In our study of flowering time, we found that using 10% bloom as a fixed threshold to define flowering time was not always reliable, as the exact moment when a tree reaches this stage can vary depending on observation conditions.

To improve data accuracy, flowering time values were adjusted based on bud development stages before and after the designated flowering date. Specifically:

- If two accessions reached 10% bloom on the same day but exhibited different bud development statuses (e.g., Figure a and b) on the previous day, their flowering time scores were adjusted (e.g., one assigned n , the other $n + 0.5$).

- If an accession continued to show approximately 10% bloom the following day, its score was further adjusted (e.g., $n+1$).

Thus, in this context, "calibrated" refers to refining flowering time data to ensure a more precise representation of phenotypic variation.

- Results:

- lines 114-117: how many SNPs in each category?

Response: The performance of the PeachSNP170K array is detailed in the table below. Additionally, we have incorporated the relevant information into the revised manuscript (lines 153-155) and

Supplementary Table 4.

“Results” section, lines 153-155:

“All samples passed quality control, with 153,431 SNPs (88.22%) meeting the quality standards; among these, 132,776 SNPs (76.34%) were classified as “PolyHighResolution”, the most stringent category (Supplementary Table 4).”

Supplementary Table 4. Performance of the PeachSNP170K array

Categorize	Number	Percentage
PolyHighResolution	132,776	76.34%
NoMinorHom	18,516	10.65%
MonoHighResolution	8,996	5.17%
Off-Target Variant	1,500	0.86%
CallRateBelowThreshold	1,993	1.15%
Other	10,144	5.83%
Total	173,925	100.00%

- line 140: remove ‘reference’

Response: The word “reference” has been removed as suggested (line 136).

- line 160: change ‘this accession’ by ‘these accessions’.

Response: The change has been made for grammatical accuracy (line 157).

- lines 178-196: Clusters should be better described. Only 201 accessions were clustered. What about the other ones? It is difficult for the reader to know which accession is in which cluster. More information (like name and origin) should be added in Supplementary data 6. Alternatively, cluster information should be moved to Supplementary Data 4.

Response: We appreciate the reviewer’s suggestion to improve the clarity of the clustering analysis. The limitation to 201 clustered accessions resulted from applying a kinship coefficient threshold (>0.45) which ensured the formation of tightly related clusters.

Of the 489 accessions analyzed:

- 201 were grouped into 25 kinship clusters (Clr1–Clr25) based on this threshold.
- 188 accessions exhibited lower kinship values and did not form any clusters.
- 100 accessions were connected as isolated pairs or small groups but lacked sufficient kinship to form cohesive clusters.

To improve clarity for the reader, we have made the following revisions:

1. We have explicitly described the distribution of clustered and unclustered accessions. The revised text reads:

“Results” section, lines 184-188:

“Of the 489 peach accessions analyzed, 382 exhibited kinship coefficients above 0.45 (Supplementary Data 5). Among these, 201 accessions were grouped into 25 multi-member kinship clusters, designated as Clr1 to Clr25 (Fig. 3d, Supplementary Data 6). The remaining accessions consisted of 188 with lower kinship values and 100 forming isolated pairs or small groups that did not meet the clustering criteria.”

2. As per your recommendation, we have provided the cluster information for each accession in an additional column in Supplementary Data 4 (489 peach varieties genotyped with the PeachSNP170K array). This adjustment improved the accessibility and interpretability of the clustering analysis.

- line 253: Associations should be clarified. ‘We identified 1207 SNPs associated...’. Perhaps this information should be summarized in regions and specifying how many regions associated per trait. Supplementary data 9 should be summarized to make it more easy to understand to the reader.

Response: Thank you for your insightful comment. We have reorganized Supplementary Data 9 to present SNP-trait associations by genomic regions and traits rather than listing individual SNPs. This restructuring provides a clearer overview of the associated genomic regions for each trait.

- line 264: Ad a dot after parenthesis and then ad ‘These findings...’.

Response: We have made the revisions in accordance with your suggestion (line 290).

- line 279: remove ‘that’

Response: The word “that” has been removed (line 304).

- line 281: remove ‘interestingly’

Response: The word "interestingly" has been removed (line 306)

- line 284: ‘whitin this block’. How many genes are annotated in this block in addition to NHX?

Response: A total of 21 genes were identified in this LD block. We have added this information to the manuscript (line 309). Among these genes, *Pp.LH.08G01962* exhibited a citrate transporter-like domain, making it a particularly strong candidate for further research due to its functional relevance.

Gene ID	Pfam ID	Interpro description
Pp.LH.08G01946	PF04199	Kynurenine formamidase(IPR007325) Kynurenine formamidase superfamily(IPR037175)
Pp.LH.08G01947	PF04199	Kynurenine formamidase(IPR007325) Kynurenine formamidase superfamily(IPR037175)
Pp.LH.08G01948	PF04199	Kynurenine formamidase(IPR007325) Kynurenine formamidase superfamily(IPR037175)
Pp.LH.08G01949	PF04199	Kynurenine formamidase(IPR007325) Kynurenine formamidase superfamily(IPR037175)
Pp.LH.08G01950	PF04199	Kynurenine formamidase(IPR007325) Kynurenine formamidase superfamily(IPR037175)

Pp.LH.08G01951	PF00560 PF13855	Leucine-rich repeat(IPR001611) Leucine-rich repeat domain superfamily(IPR032675)
Pp.LH.08G01952	NA	Ubiquitin-like domain superfamily(IPR029071)
Pp.LH.08G01953	PF00240	Ubiquitin domain(IPR000626) Ubiquitin-like domain superfamily(IPR029071)
Pp.LH.08G01954	PF00240	Ubiquitin domain(IPR000626) Ubiquitin-like domain superfamily(IPR029071)
Pp.LH.08G01955	PF00240	Ubiquitin domain(IPR000626) Ubiquitin-like domain superfamily(IPR029071)
Pp.LH.08G01956	PF00240	Ubiquitin domain(IPR000626) Ubiquitin-like domain superfamily(IPR029071)
Pp.LH.08G01957	PF00240	Ubiquitin domain(IPR000626) Ubiquitin-like domain superfamily(IPR029071)
Pp.LH.08G01958	PF00240	Ubiquitin domain(IPR000626) Ubiquitin-like domain superfamily(IPR029071)
Pp.LH.08G01959	PF00240	Ubiquitin domain(IPR000626) Ubiquitin-like domain superfamily(IPR029071)
Pp.LH.08G01960	PF00025	Small GTP-binding protein domain(IPR005225) Small GTPase superfamily, ARF/SAR type(IPR006689) P-loop containing nucleoside triphosphate hydrolase(IPR027417)
Pp.LH.08G01961	NA	NA
Pp.LH.08G01962	PF03600	Citrate transporter-like domain(IPR004680)
Pp.LH.08G01963	PF00389 PF01842 PF02826	ACT domain(IPR002912) D-isomer specific 2-hydroxyacid dehydrogenase, catalytic domain(IPR006139) D-isomer specific 2-hydroxyacid dehydrogenase, NAD-binding domain(IPR006140) D-3-phosphoglycerate dehydrogenase(IPR006236) S-adenosyl-L-homocysteine hydrolase, NAD binding domain(IPR015878) Allosteric substrate binding domain superfamily(IPR029009) D-isomer specific 2-hydroxyacid dehydrogenase, NAD-binding domain conserved site 1(IPR029752) D-isomer specific 2-hydroxyacid dehydrogenase, NAD-binding domain conserved site(IPR029753) NAD(P)-binding domain superfamily(IPR036291)
Pp.LH.08G01964	PF03629	Sialate O-acetylerase domain(IPR005181) SGNH hydrolase superfamily(IPR036514)
Pp.LH.08G01965	PF00069 PF00560 PF08263 PF13855	Protein kinase domain(IPR000719) Leucine-rich repeat(IPR001611) Protein kinase-like domain superfamily(IPR011009) Leucine-rich repeat-containing N-terminal, plant-type(IPR013210) Leucine-rich repeat domain superfamily(IPR032675)
Pp.LH.08G01966	PF12263	Protein of unknown function DUF3611(IPR022051)

Abbreviations: NA, not applicable.

- line 321: change ‘alternative’ by ‘different’

Response: The word “alternative” has been replaced with "different" for improved clarity (line 346).

- line 323: I do not think that studies on peach flowering time are limited, and previous knowledge should be summarized here or in the introduction.

Response: Thank you for your feedback. We have expanded the introduction to provide a more comprehensive summary of previous studies on peach flowering time, highlighting findings related to the *Evg* locus and its role in dormancy and flower regulation (lines 78-81).

“Introduction” section, line 78-81:

“Flowering time is crucial for the survival and reproduction in peach. Several loci and genes regulating bud dormancy and flowering time have been identified^{18,48,49}, including the six tandemly repeated *Dormancy-Associated MADS-box (DAM 1-6)* genes and their regulators⁵⁰⁻⁵³.”

- line 328: is this block close to the position where the *Evg* peach gene has been described (Bielenberg et al. 2004; <https://doi.org/10.1093/jhered/esh057>)?

Response: Thank you for your comments. We have compared the genomic position of the *Evg* locus with the flowering time-associated region identified in our study. The *Evg* locus (Chr1: 46,359,948-46,423,013) is approximately 100 kb away from our identified region (Chr1: 46,205,972-46,264,046).

To improve clarity, we have expanded the “Discussion” section (line 436-445) to provide additional background on DAM genes within the *Evg* locus, which are associated with bud dormancy and chilling requirements. While these genes are needed for perennial plant development, flowering time regulation is influenced by multiple pathways, including autonomous, photoperiod, vernalization, and thermosensory mechanisms.

Notable, we identified a *CRY* gene within the flowering time-associated region, which is known to regulate thermosensory flowering in *Arabidopsis thaliana*. Given the apparent sensitivity of this locus to local temperature conditions, we now discuss *CRY* as a potential candidate gene contributing to flowering time variation.

We hope these additions provide a clearer context for interpreting our findings.

“Discussion” section, lines 436-445:

“A major locus on chromosome 1 associated with flowering time was also identified. Candidate genes at this locus included *Pp.LH.01G06402* and *Pp.LH.01G06403*, encoding MED and CRY proteins, respectively. This locus was in close proximity to the well-established flowering time locus *Evergrowing (EVG)*, which contains six DAM genes⁵⁰. These genes are located within the genomic region Chr1: 46,205,972-46,264,046 bp, approximately 100 kb away from the major loci identified in our study. Flowering time regulation is complex, involving autonomous, photoperiod, vernalization, and gibberellin pathways⁷⁰. In woody plants, it is also linked to bud dormancy and chilling requirements^{71,72}. While DAM genes are associated with bud dormancy^{51,53,73}, the CRY gene likely plays a critical role in thermosensory flowering, as its homolog mediates this pathway in *Arabidopsis thaliana*⁷⁴.”

- line 345: are the 20 SNPs necessary to assess flowering time? Please further explain.

Response: Thank you for raising this important question. To refine our SNP selection, we conducted further analysis and optimized the identification method for early-flowering peaches. The following steps summarize our approach:

1. **Sample Selection:** We selected 35 samples with genotypes of R (reference allele)/A (alternate allele) or A/A in at least 12 of the 20 SNPs and constructed a reference pool.
2. **SNP Reduction:** By systematically testing different SNP combinations, we identified a minimal but effective set of 12 SNPs that maintained high predictive accuracy.

3. **Further Optimization:** After testing adjacent SNPs and removing those lower sample pass rates, we reduced the set to 7 SNPs while maintaining an accuracy of 93.56% (2019) and 88.94% (2020).
4. **Final Selection:** A cost-effective subset of 3 SNPs (AX-159005763, AX-158794540 and AX-159005803) were identified, achieving an accuracy of 93.97% (2019) and 88.94% (2020) using a threshold of at least two SNPs matches for early flowering classification.

These refinements have been incorporated into the “Method” section (lines 574-580).

“Methods” section, lines 574-580:

“Based on the flanking sequences of 20 SNPs comprising the early-flowering haplotype (Hap3), a set of KASP markers consisting of three SNPs was developed (Supplementary Table 8). These markers were selected based on phenotypic accuracy, genotyping performance, sample pass rate, and cost-effectiveness, specifically for early-flowering peach selection. If an SNP genotype aligned with the reference allele, it was designated as “R”; otherwise, it was designated as “A” (Alternate). When two or more of the three SNPs exhibited an “R/A” or “A/A” genotype, the sample was recognized as early-flowering.”

Reviewer #2 (Remarks to the Author):

In this article, Xu et al. report the creation of a high-throughput Axiom genotyping array containing 170K SNPs. The SNPs included in the array were identified in the resequencing of 96 peach accessions through a two-step approach, selecting only "PolyHighRes" and "NoMinorHomo" SNPs in an intermediate array containing 620K SNPs. The SNPs from the array developed by the International Peach SNP Consortium (IPSC) were also included in the final array. Xu et al. use the created array to perform an association study with 11 traits related to quality and adaptation, and to attempt the reconstruction of the relationships of nearly 500 peach accessions from the National Peach Germplasm Repository of China.

The paper appears as clear at a first glance, but several issues remain, especially considering the large amount of results presented. In my modest opinion there is material for at least two papers. In some sections, it is evident that the text was produced by different authors and therefore an effort to harmonize the writing would make the article more readable.

Response: Thank you for your valuable feedback. We appreciate your insights on improving the clarity and coherence of our manuscript. Following your guidance, we have engaged a professional editing service to refine the writing style and ensure a more unified presentation across all sections, enhancing the readability and consistency of our revised manuscript. While we acknowledge that our study presents a substantial amount of results, we believe they collectively contribute to a comprehensive understanding of peach genetic diversity and key agronomic traits.

In the presentation of the results of the final array, it is not clear how many of the 170K markers work in the final population and how they are classified into the different classes. Furthermore, it is not clear whether the array will be available through ThermoFisher or other channels or if it will remain private.

Response: Thanks for your comments. We have now explicitly state that 61,425 SNPs were used for GWAS and haplotype analyses (Method, line 547). Additionally, we have included a detailed performance assessment of the PeachSNP170K array in Supplementary Table 4 and clarified the SNP classification the revised text (line 153-155).

"Results" section, lines 153-155:

"All samples passed quality control, with 153,431 SNPs (88.22%) meeting the quality standards; among these, 132,776 SNPs (76.34%) were classified as "PolyHighResolution", the most stringent category (Supplementary Table 4)."

Supplementary Table 4. Performance of the PeachSNP170K array

Categorize	Number	Percentage
PolyHighResolution	132,776	76.34%
NoMinorHom	18,516	10.65%
MonoHighResolution	8,996	5.17%
Off-Target Variant	1,500	0.86%
CallRateBelowThreshold	1,993	1.15%

Other	10,144	5.83%
Total	173,925	100.00%

Through our validation of the array, we will make the SNP information publicly available alongside this paper. This will enable its use for research and breeding purposes through solid-phase arrays such as those provided by ThermoFisher Scientific, as well as liquid-phase arrays. By sharing this information, we also aim to facilitate further array development, offering greater flexibility to researchers and breeders for diverse applications in different scenarios.

The introduction is clear, I only have a couple of minor revisions:

- lines 40-41: the adjective "recently" does not seem appropriate for a reference from 2012 or 2020.

Response: Thank you for your suggestion. We have replaced the word "recently" with "in 2020" to accurately reflect when the 18K chip was reported (line 41).

- line 57: the link to the website "rosaceae.org" is too general. This website is a big repository of many different types of data, not only of peaches.

Response: Thank you for your suggestion. We have replaced the general link with a direct link to [Peach Genomic and transcriptomic Resources \(https://www.rosaceae.org/organism/24333?pane=bio_data_1_rsc_genomes\)](https://www.rosaceae.org/organism/24333?pane=bio_data_1_rsc_genomes) (line 56-57).

In the results, it is necessary to add details on the performance of the final array. The development of the intermediate array from the resequencing is clearly and appropriately described, but details are missing on the classification of the final SNPs in the 489 accessions used for the association study and for the reconstruction of the parental relationships.

- line 114: verify "MonoHighResolution", in other parts of the article it talks about "NoMinorHom".
Supplementary figure 1: check the plot labels, some of the classes are mislabeled.

Response: Thank you for your careful review and for pointing out the inconsistency in our manuscript. We have corrected "MonoHighResolution" with "NoMinorHom" at line 111 of the revised manuscript. Additionally, we have updated the typical SNP category patterns for "NoMinorHom" and "off-target variant" with the correct and updated representations in Supplementary Fig. 1.

Supplementary Fig. 1 Cluster plot examples of the SNP classification categories based on the call derived from genotyping of the 192 peach accessions.

Classification categories are assigned by R package SNPolisher (Nicolazzi et al., 2014) according to the Axiom™ Genotyping Solution Data Analysis user guide, into six major types: “PolyHighResolution”, “MonoHighResolution”, “NoMinorHomozygote”, “Off-Target Variant”, “CallRateBelowThreshold,” or “Other”.

- lines 113-117: it is not clear where the total of 173,925 SNPs comes from considering 132,776 + 18,561 + 9K.

Response: Thank you for your careful review. The numbers 132,776 and 18,561 SNPs in the original manuscript correspond to the “PolyHighResolution” and “NoMinorHom” categories, respectively, when assessing the PeachSNP170K array. However, for the development of the PeachSNP170K array, the correct counts for these categories are 98,884 SNPs for “PolyHighResolution” and 90,684 SNPs for “NoMinorHorm” based on the performance of the 620K intermediate array. This correction has been implemented in lines 110-115 of the revised manuscript.

“Results” section, lines 110-115:

“Only the SNPs from the two highest-priority categories— “PolyHighResolution” (98,884 SNPs) and “NoMinorHorm” (90,684 SNPs) (Supplementary Fig. 1)—were retained. SNPs with a call rate below 97.5% were removed, leaving 166,416 SNPs. Additionally, 8,996 SNPs from the IPSC peach 9K SNP array¹³ were integrated, while 1,487 overlapping SNPs were excluded. This progress resulted in a final set of 173,925 SNP sites for constructing the PeachSNP170K array (Fig. 1; Supplementary Table 1)”

- lines 128-129: the reference to PyrSNParray seems inappropriate. Verify if it is a typo otherwise remove it or move the comparison with other arrays to the discussion to avoid confusion.

Response: We have carefully reviewed the reference to PyrSNParray and confirmed that it is not a typo but an intended comparison with the peach170KSNP array. However, we acknowledge the

potential for confusion. In light of your comments, we have decided to remove the reference to the PyrSNParray from the manuscript (line 126).

In the kinship study, it is not clear whether the authors consider the high rate of inbreeding and self-pollination that is present in *Prunus*? The use of the kinship value (>0.45) alone, without considering the number of loci that present potential Mendelian errors ($IBD = 0$), could lead to a very high number of false positives in the identified pedigrees.

Response: We sincerely appreciate the reviewer's insightful comments regarding inbreeding, self-pollination in *Prunus* species, and kinship values interpretation. In response, we have reanalyzed the inbreeding coefficients (F-values) based on the updated dataset, yielding the following results:

Peach landraces exhibited higher inbreeding levels than cultivated peaches, with a mean F-value of 0.345 (± 0.240). This suggests that self-compatibility, along with long-term selection and purification, has contributed to increased homozygosity.

- Peach cultivars displayed lower and more variable inbreeding levels, with F-values ranging from 0.075 to 0.111, depending on geographic origin. Cultivars from America (0.111) had slightly higher inbreeding levels compared to those from Europe (0.075), China (0.078), and other Asian countries (0.098). This variation likely reflects differences in breeding strategies, hybridization histories, and selection pressures across regions.
- Compared to fully self-pollinating crops such as rice and soybean ($F \approx 0.90-0.99$), *Prunus* species exhibit much lower inbreeding levels, indicating that despite self-compatibility, gene flow and hybridization have prevented extreme inbreeding in cultivated peaches.

These findings have been incorporated into the revised manuscript (lines 176-181), and a detailed breakdown of F-values by geographic group is now available in Supplementary Table 5.

“Results section”, lines 176-181:

“To further investigate genetic relatedness and inbreeding within cultivated peach, including both landraces and cultivars, we analyzed inbreeding coefficients (F-values) (Supplementary Table 5). Landraces exhibited moderate to high inbreeding levels (mean: 0.344 to 0.531), likely due to self-compatibility and long-term selection progresses. In contrast, cultivars displayed lower F-values (mean F: -0.067 to 0.111), reflecting genetic diversity introduced through natural and artificial hybridization, as well as modern breeding practices promoting greater gene flow.”

Supplementary Table 5. Inbreeding coefficients (F-values) in peach landraces and cultivars.

Peach Group	Mean F-value	Standard Deviation (SD)
European cultivars	0.075	0.229
American cultivars	0.111	0.184
Landraces	0.345	0.240
Chinese cultivars	0.078	0.148

Additionally, we acknowledge that our previous description may have caused some misunderstandings regarding the scope of our kinship study. Our goal is to use SNP-based kinship values as a genetic reference framework rather than to reconstruct detailed pedigrees. We fully agree with that if the objective were pedigree reconstruction, accounting for IBD = 0 loci would be essential to minimize false positives. We have clarified this in the Discussion section (lines 412-417).

“Discussion” section, lines 412-417:

“SNP-based kinship analysis provides a genome-wide perspective on genetic relationships and serves as a complementary tool to pedigree data rather than a direct substitute. While our findings demonstrate the utility of SNP-based kinship analysis in assessing genetic relationships, we recognize its limitations, such as the potential impact of Mendelian errors (e.g., IBD = 0) on kinship estimates. Since our study focuses on establishing a genetic reference framework rather than reconstructing pedigrees, these considerations were not incorporated into our analysis.”

In the Methods section and particularly in the subsections “SNP selection for the intermediate 620K SNP array” and “Final SNP selection for construction of the PeachSNP170K array” there is a mix of Results and Methods.

Response: We have revised the subsections to ensure a clear distinction between Methods and Results (lines 97-115 and 457-508).

At row 503 there is a section about Selective sweeps but the results of this analysis are not presented and discussed, they are only briefly cited at row 287. The results of this analysis should be better discussed or totally removed from the paper.

Response: We appreciate the reviewer’s observation and have expanded both the results and discussion of the selective sweep analysis to ensure they are fully integrated into the manuscript.

- Results section (lines 313-317): We have explicitly presented the selective sweep findings. We highlighted the identified loci’s evolutionary and breeding significance and potential future applications.
- Discussion section (lines 430-435): We have elaborated on the evolutionary and breeding significance of the identified loci and their potential applications in marker-assisted selection.

These revisions ensure that the selective sweep analysis is thoroughly contextualized.

“Results” section, lines 313-317:

“Strong selective signatures (XP-CLR) were observed within the *PpNHX1*-associated genomic block (Chr8: 17,154,192-17,230,631 bp) in Clr1 and Clr6 accessions (Supplementary Fig. 5). This suggests that divergent selection acted on this region, likely reflecting artificial selection preferences for citrate accumulation in these clusters. Such selection may indicate the breeding significance of *PpNHX1* in shaping fruit acidity traits in peach.”

“Discussion” section, lines 430-435:

“The integration of selective sweep analysis provides additional insights into the evolutionary and breeding relevance of key loci associated with fruit acidity. Divergent selection between Clr1 (low-citrate content) and Clr6 (high-citrate content) accessions likely reflects artificial selection preferences for citrate accumulation. These findings complement the GWAS results, reinforcing the role of *PpNHX1* as a key determinant of fruit acidity and offering valuable insights for marker-assisted selection strategies targeting this trait.”